# Accurate and interpretable drug-drug interaction prediction enabled by knowledge subgraph learning
Yaqing Wang [1,4], Zaifei Yang [1,2,4] & Quanming Yao[3] ✉

## Abstract

**Background** Discovering potential drug-drug interactions (DDIs) is a long-standing challenge in clinical treatments and drug developments. Recently, deep learning techniques have been developed for DDI prediction. However, they generally require a huge number of samples, while known DDIs are rare.

**Methods** In this work, we present KnowDDI, a graph neural network-based method that addresses the above challenge. KnowDDI enhances drug representations by adaptively leveraging rich neighborhood information from large biomedical knowledge graphs. Then, it learns a knowledge subgraph for each drug-pair to interpret the predicted DDI, where each of the edges is associated with a connection strength indicating the importance of a known DDI or resembling strength between a drug-pair whose connection is unknown. Thus, the lack of DDIs is implicitly compensated by the enriched drug representations and propagated drug similarities.

**Results** Here we show the evaluation results of KnowDDI on two benchmark DDI datasets. Results show that KnowDDI obtains the state-of-the-art prediction performance with better interpretability. We also find that KnowDDI suffers less than existing works given a sparser knowledge graph. This indicates that the propagated drug similarities play a more important role in compensating for the lack of DDIs when the drug representations are less enriched.

**Conclusions** KnowDDI nicely combines the efficiency of deep learning techniques and the rich prior knowledge in biomedical knowledge graphs. As an original open-source tool, KnowDDI can help detect possible interactions in a broad range of relevant interaction prediction tasks, such as protein-protein interactions, drug-target interactions and disease-gene interactions, eventually promoting the development of biomedicine and healthcare.

## Plain Language Summary

Understanding how drugs interact is crucial for safe healthcare and the development of new medicines. We developed a computational tool that can analyze the data about medicines within large medical databases and predict the impact of being treated by multiple drugs at the same time on the person taking the drugs. Our tool, named KnowDDI, can predict which drugs interact with each other and also provide an explanation for why the interaction is likely to take place. We demonstrated that our tool can identify known drug interactions. It could potentially be used in the future to identify previously unknown or unanticipated interactions that could have negative consequences to people being treated with unusual combinations of medicines.

Accurately predicting drug-drug interaction (DDI) can play an important role in the field of biomedicine and healthcare. On the one hand, combination therapies, where multiple drugs are used together, can be used to treat complex disease and comorbidities, such as human immunodeficiency virus (HIV)[1,2]. Recent study also shows that combination therapies, such as a combination of lopinavir and ritonavir, may treat coronavirus disease (COVID-19)[3–5], the infectious disease which causes global pandemic in the past three years. On the other hand, DDI is an important cause of adverse drug reactions, which accounts for 1% hospitalizations in the general population and 2–5% hospital admissions in the elderly[6–8]. A concrete example is that if warfarin and aspirin enter the body together, they will compete for binding to plasma proteins. Then, the remained warfarin that cannot be bounded to plasma proteins will remain in the blood, which results in acute bleeding in patients[9].

Identifying DDIs by clinical evidence such as laboratory studies is extremely costly and time-consuming[6,8]. In recent years, computational techniques especially deep learning approaches are developed to speed up the discovery of potential DDIs. Naturally, DDI fact triplets can be represented as a graph where each node corresponds to a drug, and each edge represents an interaction between two drugs. Provided with DDI fact

[1]Baidu Research, Baidu Inc., Beijing, China. [2]Institute of Computing Technology, Chinese Academy of Sciences, Beijing, China. [3]Department of Electronic Engineering, Tsinghua University, Beijing, China. [4]These authors contributed equally: Yaqing Wang, Zaifei Yang. ✉e-mail: qyaoaa@tsinghua.edu.cn

triplets, a number of graph learning methods have been developed to identify unknown interactions between drug-pairs. Graph neural networks (GNNs)[10,11], which can obtain expressive node embeddings by end-to-end learning from the topological structure and associated node features, have also been applied for DDI prediction problem. However, known DDI fact triplets are rare due to the high experimental cost and continually emerging new drugs[12]. For example, the latest DrugBank database with 14,931 drug entries only contains 365,984 known DDI fact triplets[13], the quantity of which is less than 1% of the total potential DDIs. This makes over-parameterized deep learning models fail to give full play to its expressive ability and may perform even worse than traditional two-stage embedding methods[14,15].

In biomedicine and healthcare, many international level agencies such as National Center for Biotechnology Information and European Bioinformatics Institute are endeavored to regularly maintain rich publicly available biomedical data resources[16]. Researchers then integrate these disparate and heterogeneous data resources into knowledge graphs (KGs) to facilitate an organized use of information. Examples are Hetionet[17,18], PharmKG[19] and PrimeKG[20]. These KGs contain rich prior knowledge discovered in biomedicine and healthcare. A proper usage of them may compensate for the lack of samples for DDI prediction. The pioneer work KGNN[21] firstly leverages external KGs to provide topological information for each drug in target drug-pair. In particular, it uniformly samples a fixed size set of neighbors around each drug, then aggregates drug features and messages from the sampled neighbors into the drug representation without considering which drug to interact. Later works merge the DDI network with external KGs as a combined network, extract enclosing subgraphs for different drug-pairs to encode the drug-pair specific information, and then predict DDI for the target drug-pair using the concatenation of nodes embeddings of drugs and subgraph embedding of enclosing subgraphs[22–24]. However, as these KGs integrate diverse data resources by automated process or experts, existing methods fail to filter out noise or inconsistent information. As a result, properly leveraging external KGs is still a challenging problem.

In this paper, we propose KnowDDI, an accurate and interpretable method for DDI prediction. First, we merge the provided DDI graph and an external KG into a combined network, upon which generic representations

for all nodes are learned to encode the generic knowledge. Next, we extract a drug-flow subgraph for each drug-pair from the combined network. We then learn a knowledge subgraph from generic representations and the drug-flow subgraph. After optimization, the representations of drugs are transformed to be more predictive of the DDI types between the target drug-pair. In addition, the returned knowledge subgraph contains explaining paths to interpret the prediction result for the drug-pair, where the explaining paths consist of only edges of important known DDIs or newly-added edges connecting highly similar drugs. In other words, the learned knowledge subgraph helps filter out irrelevant information and adds in resembling relationships between drugs whose interactions are unknown. This allows the lack of DDIs to be implicitly compensated by the enriched drug representations and propagated drug similarities. We perform extensive experimental results on benchmark datasets, and observe that KnowDDI consistently outperforms existing works. We also conduct a series of case studies which further show that KnowDDI can discover convincing explaining paths which help interpret the DDI prediction results. KnowDDI has the potential to be used in a broad range of relevant interaction prediction tasks, such as protein-protein interactions, drug-target interactions and disease-gene interactions to help detect potential interactions, eventually advancing the development of biomedicine and healthcare.

## Methods
### Overview of KnowDDI

Our KnowDDI (Fig. 1) learns to predict DDIs between a drug-pair, i.e., head drug $h$ and tail drug $t$ in the DDI graph, by learning with knowledge subgraph, i.e., denoted as $\mathcal{S}_{h,t}$. The provided DDI graph and an external KG are merged into a combined network as the start. Every node of the combined network is associated with a unique generic embedding which is learned to encode the generic knowledge. Given a target drug-pair $(h, t)$, a drug-flow subgraph $\bar{\mathcal{S}}_{h,t}$ which captures local context relevant to $(h, t)$ is extracted from the combined network. As directly leveraging external KG (and hence $\bar{\mathcal{S}}_{h,t}$) may bring in irrelevant information, the graph structure and node embeddings of $\bar{\mathcal{S}}_{h,t}$ are further iteratively optimized. During this process, the generic embeddings are transformed to be more predictive of the DDI types between the target drug-pair. In addition, KnowDDI estimates a connection

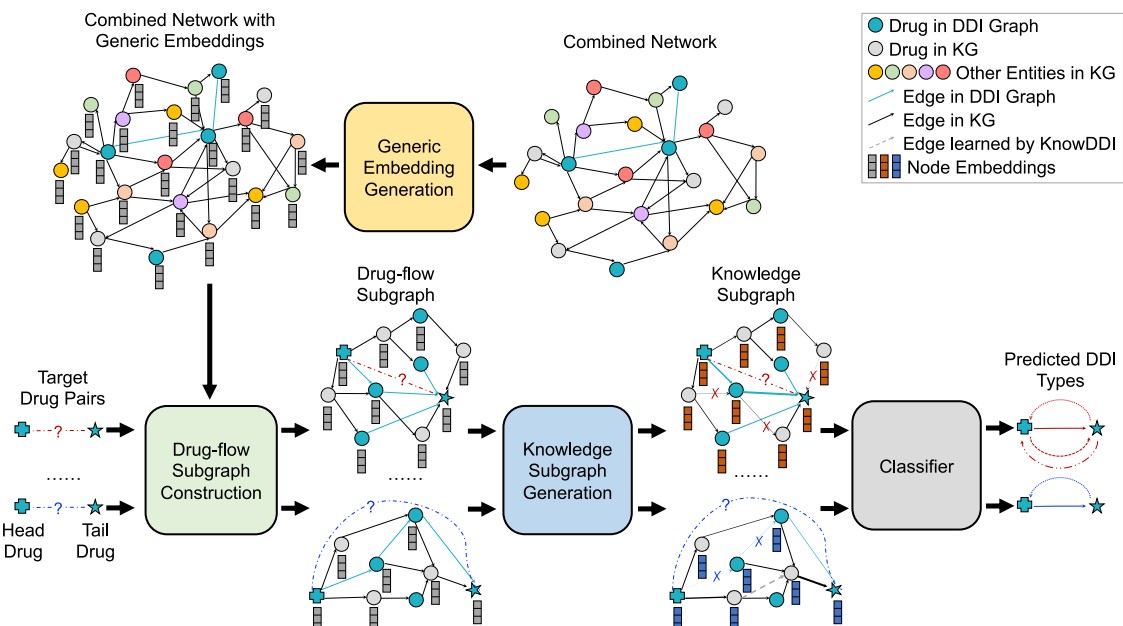

**Fig. 1 | Overview of KnowDDI.** On a combined network which merges the drug-drug interaction (DDI) graph with an external knowledge graph (KG), generic embeddings of all nodes are firstly learned to capture generic knowledge. Then for each target drug-pair, a drug-flow subgraph is extracted from the combined network, whose node embeddings are initialized as the generic embeddings. Via propagating drug resembling relationships, the generic embeddings are transformed to be more predictive of the DDI types between the drug-pair, and the drug-flow subgraph is adapted as knowledge subgraph which contains explaining paths to interpret the prediction result.

strength for every drug-pair in the subgraph, representing the importance of a given edge between connected nodes in the drug-flow subgraph or similarity between two nodes whose connection is unknown. Accordingly, a new edge of type "resemble" is added between two nodes if their node embeddings are highly similar, and existing edges can be dropped if the importance is estimated as low. Thus, only useful information flows between nodes are kept. The final optimized subgraph becomes our knowledge subgraph $\mathcal{S}_{h,t}$ which consists of explaining paths. The average connection strength over all consecutive node-pairs along each explaining path indicates its ability of explaining the current prediction result in the perspective of KnowDDI. Supplementary Table 1 shows a summary of characteristics comparing KnowDDI with existing works.

## Problem setup

A *Drug-Drug Interaction (DDI) graph* is denoted as $\mathcal{G}_{\mathrm{DDI}} = \{\mathcal{V}_{\mathrm{DDI}}, \mathcal{E}_{\mathrm{DDI}}, \mathcal{R}_{\mathrm{DDI}}\}$, where $\mathcal{V}_{\mathrm{DDI}}$ is a set of drug nodes, $\mathcal{E}_{\mathrm{DDI}}$ is a set of edges, and $\mathcal{R}_{\mathrm{DDI}}$ is a set of DDI relation types associated with the edges. In particular, each edge $(u, r, v) \in \mathcal{E}_{\mathrm{DDI}}$ corresponds to an observed fact triplet, which records the DDI relation type $r \in \mathcal{R}_{\mathrm{DDI}}$ associated with $(u, v)$.

The external *Knowledge Graph (KG)* is denoted as $\mathcal{G}_{\mathrm{KG}} = \{\mathcal{V}_{\mathrm{KG}}, \mathcal{E}_{\mathrm{KG}}, \mathcal{R}_{\mathrm{KG}}\}$, which contains rich biomedical knowledge of various kinds of biomedical entities. Particularly, $\mathcal{V}_{\mathrm{KG}}$ consists of $|\mathcal{V}_{\mathrm{KG}}|$ entities ranging from drugs, genes to proteins, $\mathcal{R}_{\mathrm{KG}}$ consists of $|\mathcal{R}_{\mathrm{KG}}|$ types of interactions occurred in $\mathcal{G}_{\mathrm{KG}}$, while $\mathcal{E}_{\mathrm{KG}} = \{(u, r, v)|u, v \in \mathcal{V}_{\mathrm{KG}}, r \in \mathcal{R}_{\mathrm{KG}}\}$ consists of observed fact triplets in $\mathcal{G}_{\mathrm{KG}}$. Usually, $\mathcal{G}_{\mathrm{KG}}$ is much larger than $\mathcal{G}_{\mathrm{DDI}}$, such that $\mathcal{V}_{\mathrm{DDI}} \subseteq \mathcal{V}_{\mathrm{KG}}$ and $\mathcal{R}_{\mathrm{DDI}} \subseteq \mathcal{R}_{\mathrm{KG}}$ hold.

The combination of $\mathcal{G}_{\mathrm{DDI}}$ and $\mathcal{G}_{\mathrm{KG}}$ then forms a large *combined network* $\mathcal{G} = \{\mathcal{V}, \mathcal{E}, \mathcal{R}\} = \{\mathcal{V}_{\mathrm{DDI}} \cup \mathcal{V}_{\mathrm{KG}}, \mathcal{E}_{\mathrm{DDI}} \cup \mathcal{E}_{\mathrm{KG}}, \mathcal{R}_{\mathrm{DDI}} \cup \mathcal{R}_{\mathrm{KG}}\}$.

The target of this paper is to learn a mapping function from the combined network $\mathcal{G}$, which can predict the relation type between new drug-pairs. For multiclass DDI prediction, each drug-pair only has one specific relation type $r \in \mathcal{R}_{\mathrm{DDI}}$. As for multilabel DDI prediction, multiple relation types $r_1, r_2, \ldots \in \mathcal{R}_{\mathrm{DDI}}$ can co-occur between a drug-pair.

## Architecture of KnowDDI

The overall architecture of KnowDDI is shown in Fig. 1. Here, we provide the details of how drug representations are enriched by an external KG and how similarities are propagated in knowledge subgraphs. The complete algorithms of training and testing KnowDDI are summarized in Supplementary Note 1.

**Generic embedding generation.** To encode the generic knowledge of various other type of entities in $\mathcal{G}$, which can help enrich representation for drug nodes, we run a GNN on the combined network $\mathcal{G}$ to obtain generic embedding of each node.

Let $\mathbf{e}_v^{(0)}$ denote the feature of node $v \in \mathcal{V}$. In KnowDDI, we follow GraphSAGE[25] and update the embedding $\mathbf{e}_v^{(l)}$ for node $v$ at the $l$th layer as:

$$\mathbf{a}_v^{(l)} = \mathrm{MEAN}\left(\{\mathrm{ReLU}(\mathbf{W}_a^{(l)} \mathbf{e}_u^{(l)}) : (u, r, v) \in \mathcal{E}\}\right), \tag{1}$$

$$\mathbf{e}_v^{(l)} = \mathbf{W}_c^{(l)} \cdot \left[\mathbf{a}_v^{(l)} \parallel \mathbf{e}_v^{(l-1)}\right], \tag{2}$$

where $\mathrm{MEAN}(\cdot)$ is element-wise mean pooling, $\mathbf{W}_a^{(l)}$ and $\mathbf{W}_c^{(l)}$ are learnable parameters, and $[\cdot \parallel \cdot]$ concatenates vectors along the last dimension. After $L$ layers of message passing, the final node embedding $\mathbf{e}_v^{(L)}$ is taken as the generic embedding of $v \in \mathcal{V}$.

**Drug-flow subgraph construction.** As each drug-pair can depend on different local contexts, i.e., entities and relations, we construct a drug-flow subgraph $\bar{\mathcal{S}}_{h,t}$ specific to $(h, t)$ from the combined network $\mathcal{G}$, which transforms drug representations obtained in Section 6 to be drug-pair-aware.

For each $(h, t)$, we define its drug-flow subgraph $\bar{\mathcal{S}}_{h,t} = \{\bar{\mathcal{V}}_{h,t}, \bar{\mathcal{E}}_{h,t}, \bar{\mathcal{R}}_{h,t}\}$ as a directed subgraph of graph $\mathcal{G}$ consisting of relational paths $\{\bar{\mathbf{p}}_{h,t}\}$ with length at most $P$ pointing from $h$ to $t$ in $\mathcal{G}$, where a relational path

$$\bar{\mathbf{p}}_{h,t} = h \xrightarrow{r_1} v_1 \xrightarrow{r_2} v_2 \cdots \xrightarrow{r_p} t, \tag{3}$$

is a sequence of nodes connected by relations. Here, $\bar{\mathcal{V}}_{h,t}$ is a set of nodes appearing in $\bar{\mathcal{S}}_{h,t}$, $\bar{\mathcal{R}}_{h,t}$ is a set of relation types occurred between nodes in $\bar{\mathcal{V}}_{h,t}$ and $\bar{\mathcal{E}}_{h,t} = \{(u, r, v)|u, v \in \mathcal{V}_{h,t} \text{ and } (u, r, v) \in \mathcal{E}\}$ is a set of edges connecting nodes in $\bar{\mathcal{V}}_{h,t}$.

Algorithm ?? in Supplementary Note 1 summarizes the procedure of extracting $\bar{\mathcal{S}}_{h,t}$. Given a drug-pair $(h, t)$, we first extract the interaction of local neighborhoods of $h$ and $t$ (i.e. $K$-hop enclosing subgraph[22], where $K$ is a hyperparameter) from $\mathcal{G}$. For computational simplicity, we make all the relational paths $\{\bar{\mathbf{p}}_{h,t}\}$ between $h$ and $t$ have length $P$. This is done by augmenting relational paths with length less than $P$ by identity relations[26,27], i.e., $(t, r_{\mathrm{identity}}, t)$. If there is no relational path connecting $h$ and $t$, we return $\bar{\mathcal{S}}_{h,t} = \{\{h, t\}, \emptyset, \emptyset\}$. As these $K$-hop enclosing subgraphs neglect directional information, we need to conduct directional pruning to remove all nodes and corresponding edges which are not on any relational path pointing from $h$ to $t$. Thus, after directional pruning, the resultant drug-flow subgraph $\bar{\mathcal{S}}_{h,t}$ only contains nodes which supports learning the information flow from $h$ to $t$.

**Knowledge subgraph generation.** Further, we learn a knowledge subgraph $\mathcal{S}_{h,t}$ from generic embeddings and the drug-flow subgraph $\bar{\mathcal{S}}_{h,t}$. During this process, irrelevant edges are removed and new edges of type "resemble" are added between nodes with highly similar node embeddings.

Let $\bar{\mathbf{A}}_{h,t}$ be a binary third-order tensor with size $|\bar{\mathcal{V}}_{h,t}| \times |\bar{\mathcal{V}}_{h,t}| \times |\bar{\mathcal{R}}_{h,t}|$. Its $(u, v, r)$th entry is computed as

$$\bar{\mathbf{A}}_{h,t}(u, v, r) = \begin{cases} 1 & \text{if } (u, r, v) \in \bar{\mathcal{E}}_{h,t} \\ 0 & \text{otherwise} \end{cases}, \tag{4}$$

which records whether drug-pair $(u, v)$ is connected by relation type $r \in \bar{\mathcal{R}}_{h,t}$ in $\bar{\mathcal{S}}_{h,t}$.

In addition, we estimate another third-order tensor $\mathbf{A}_{h,t}$ from $\bar{\mathbf{A}}_{h,t}$ with elements $\mathbf{A}_{h,t}(u, v, r) \in [0, 1]$ and size $|\bar{\mathcal{V}}_{h,t}| \times |\bar{\mathcal{V}}_{h,t}| \times (|\bar{\mathcal{R}}_{h,t}| + 1)$ to record the connection strength between nodes $u, v \in \bar{\mathcal{V}}_{h,t}$ w.r.t. relation $r$. Specifically, if $\bar{\mathbf{A}}_{h,t}(u, v, r) = 1$ but $\mathbf{A}_{h,t}(u, v, r) = 0$, this means the existing edge $(u, r, v)$ is not useful and should be removed. Besides, if $\bar{\mathbf{A}}_{h,t}(u, v, r) = 0$ for all $r \in \bar{\mathcal{R}}_{h,t}$, we add an edge of relation type "resemble" $r_{\mathrm{sim}} \in \mathcal{R}$ to connect $u$ and $v$. $\mathbf{A}_{h,t}(u, v, r_{\mathrm{sim}}) > 0$ then represents the similarity between $u$ and $v$. Corresponding to $\mathbf{A}_{h,t}$ and $\bar{\mathcal{S}}_{h,t}$, the knowledge subgraph $\mathcal{S}_{h,t}$ is generated as

$$\mathcal{S}_{h,t} = \{\bar{\mathcal{V}}_{h,t}, \mathcal{E}_{h,t}, \mathcal{R}_{h,t}\}, \tag{5}$$

where $\mathcal{R}_{h,t} = \{r_{\mathrm{sim}}\}\bar{\mathcal{R}}_{h,t}$, and $\mathcal{E}_{h,t} = \{(u, r, v)\}$ with each $(u, r, v)$ constructed as $(u, r, v) \in \bar{\mathcal{E}}_{h,t}$ if $\bar{\mathbf{A}}_{h,t}(u, v, r) = 1 \wedge \mathbf{A}_{h,t}(u, v, r) > 0$, or $(u, r_{\mathrm{sim}}, v)$ if $\bar{\mathbf{A}}_{h,t}(u, v, r) = 0 \wedge \mathbf{A}_{h,t}(u, v, r_{\mathrm{sim}}) > 0$.

To learn such a $\mathbf{A}_{h,t}$, we conduct graph structure learning to alternate the following two steps for $T$ times:
- estimate connection strengths between every pair of nodes in $\bar{\mathcal{V}}_{h,t}$, and
- refine node embeddings on the updated subgraph.

First, we initialize the node embedding of each $v \in \bar{\mathcal{V}}_{h,t}$ as $\mathbf{h}_u^{(0)} = \mathbf{e}_u^{(L)}$ to encode the global topology of $\mathcal{G}$. Let $\mathbf{A}_{h,t}^{(\tau)}$ be the estimation of $\mathbf{A}_{h,t}$ at the $\tau$th iteration. We initialize $\mathbf{A}_{h,t}^{(0)} = \bar{\mathbf{A}}_{h,t}$. Next, we estimate relevance score $\mathbf{C}_{h,t}^{(\tau)}(u, v, r)$ for each relation $r \in \mathcal{R}_{h,t}$ between every node-pair $(u, v)$ in $\mathcal{S}_{h,t}$ as

$$\mathbf{C}_{h,t}^{(\tau)}(u, v, r) = \mathrm{MLP}\left(\left[\mathbf{h}_{uv}^{\tau-1} \parallel \mathbf{h}_r\right]\right), \tag{6}$$

where $\mathbf{h}_{uv}^{\tau-1} = \exp(-|\mathbf{h}_u^{(\tau-1)} - \mathbf{h}_v^{(\tau-1)}|)$, $|\cdot|$ returns the element-wise absolute value, MLP is multi-layer perception, and $\mathbf{h}_r$ is the learnable relation embedding of relation $r$. We set $\mathbf{C}_{h,t}^{(\tau)}(v, v, r) = 1$ for all $v \in \bar{\mathcal{V}}_{h,t}$ and $r \in \mathcal{R}_{h,t}$.

This learned $\mathbf{C}_{h,t}^{(\tau)}$ reveals how the model understands the connection between different node-pairs at the $\tau$th iteration. It helps filter out irrelevant information and captures resembling relationships between drugs whose interactions are unknown. In the early stage of optimization, $\mathbf{C}_{h,t}^{(\tau)}$ can be less trustworthy. Hence, we merge this learned subgraph with $\bar{\mathbf{A}}_{h,t}$ to obtain $\mathbf{A}_{h,t}^{(\tau)}$, i.e.,

$$\mathbf{A}_{h,t}^{(\tau)} = \mathrm{ReLU}\left(\delta\left(\alpha\mathbf{A}_{h,t}^{(0)} + (1-\alpha)\mathbf{C}_{h,t}^{(\tau)}\right) - \gamma\right), \tag{7}$$

where hyperparameter $\alpha$ is used to balance their contribution in the final prediction, the threshold $\gamma \geq 0$ is used to screen out those less informative edges, and $[\mathrm{ReLU}(x)] = \max(x, 0)$. Considering that nodes are connected by different numbers of neighbors, we use function $\delta(\cdot)$ to ensure that the relevance scores of incoming edges for $v$ sum into 1, i.e.,

$$\sum_{i \in \bar{\mathcal{V}}_{h,t}} \sum_{r \in \mathcal{R}_{h,t}} \mathbf{A}_{h,t}^{(\tau)}(i, v, r) = 1. \tag{8}$$

Here, we instantiate $\delta(\cdot)$ as edge softmax function which computes softmax over attention weights of incoming edges regardless of their relation types for every node, i.e., $\mathrm{softmax}(x_i) = \exp(x_i)/\sum_{j \in \mathcal{N}(i)} \exp(x_j)$ where $\mathcal{N}(i) = \{j : (j, r, i) \in \mathcal{E}_{h,t}\}$. Let $\mathbf{H}_{h,t}^{(\tau)}$ be embeddings of all nodes in $\bar{\mathcal{V}}_{h,t}$ where the $v$th row corresponds to node embedding $\mathbf{h}_v^{(\tau)}$ of $v \in \bar{\mathcal{V}}_{h,t}$, and

$$\mathbf{Q}_{h,t,r}^{(\tau)} = \mathrm{ReLU}\left(\mathbf{A}_{h,t}^{(\tau)}(:, :, r)\mathbf{H}_{h,t}^{(\tau-1)}\mathbf{W}_r\right), \tag{9}$$

where $\mathbf{A}_{h,t}^{(\tau)}(:, :, r)$ is the $r$th slice of $\mathbf{A}_{h,t}^{(\tau)}$ and $\mathbf{W}_r$ is a learnable parameter. Then, $\mathbf{H}_{h,t}^{(\tau)}$ is updated as

$$\mathbf{H}_{h,t}^{(\tau)} = \mathrm{MEAN}(\{\mathbf{Q}_{h,t,r}^{(\tau)} : r \in \mathcal{R}_{h,t}\}). \tag{10}$$

After $T$ iterations, the representations of drugs are transformed to be more predictive of the DDI types between the target drug-pair, and $\mathcal{S}_{h,t}$ only keeps edges of important known DDIs or newly-added edges connecting highly similar drugs. We set $\mathbf{h}_v = \mathbf{h}_v^{(T)}$ as the final node embedding, and return $\mathbf{A}_{h,t} = \mathbf{A}_{h,t}^{(T)}$ which records the updated graph structure of $\mathcal{S}_{h,t}$. Learning subgraph embedding of $\mathcal{S}_{h,t}$ is commonly adopted to encode the subgraph topology [22,23,25]. Hence, we follow this routine and obtain the subgraph embedding $\mathbf{h}_{\mathcal{S}_{h,t}}$ of $\mathcal{S}_{h,t}$ as

$$\mathbf{h}_{\mathcal{S}_{h,t}} = \mathrm{MEAN}\left(\{\mathbf{h}_v | v \in \bar{\mathcal{V}}_{h,t}\}\right). \tag{11}$$

Finally, we predict the relation for $(h, t)$ as

$$\hat{\mathbf{y}}_{h,t} = \delta\left(\mathbf{W}_c \cdot \left[\mathbf{h}_{\mathcal{S}_{h,t}} \| \mathbf{h}_h \| \mathbf{h}_t\right]\right), \tag{12}$$

where $\mathbf{W}_c$ is the classifier parameter.

The results of applying different knowledge subgraph generation strategies are shown in Supplementary Fig. 1 and analyzed in Supplementary Note 2.

**Learning and inference.** Let $\boldsymbol{\theta}_g$ and $\boldsymbol{\theta}_k$ denote the collection of parameters associated with generic embedding generation and knowledge subgraph generation respectively. Further, let $\mathbf{y}_{h,t} = [\mathbf{y}_{h,t}(i)]$ be a vector where the $i$th element $\mathbf{y}_{h,t}(i) = 1$ if relation $i \in \mathcal{R}_{\mathrm{DDI}}$ occurs in $(h, t)$ and 0 otherwise.

For multiclass DDI prediction, we optimize KnowDDI w.r.t. the cross entropy loss:

$$\min_{\boldsymbol{\theta}_g, \boldsymbol{\theta}_k} \ell_{\mathrm{CE}} \equiv \sum_{(h,r,t) \in \mathcal{E}_{\mathrm{DDI}}} -\mathbf{y}_{h,t}^\top \cdot \log\left(\hat{\mathbf{y}}_{h,t}\right). \tag{13}$$

As for multilabel DDI prediction, drug-pairs are associated with varying number of relations. We further use a loss function with negative sampling. Following related works[22,23], we construct negative triplets to prevent KnowDDI from selecting those unknown relations. For each $(h, r, t) \in \mathcal{E}_{\mathrm{DDI}}$, we replace $t$ by a randomly sampled drug $w \in \mathcal{V}_{\mathrm{DDI}}$ to form $(h, r, w)$ whose label vector $\mathbf{y}_{h,w} = [0, \ldots, 0]$ contains zeros only. Let $\mathcal{E}_{\mathrm{neg}} = \{(h, r, w) | (h, r, t) \in \mathcal{E}_{\mathrm{DDI}} \text{ and } (h, r, w) \notin \mathcal{E}_{\mathrm{DDI}}\}$ collectively contains the negative triplets. We optimize KnowDDI w.r.t. the following loss for multilabel DDI prediction:

$$\min_{\boldsymbol{\theta}_g, \boldsymbol{\theta}_k} \ell_{\mathrm{CE}} + \sum_{(h,r,w) \in \mathcal{E}_{\mathrm{neg}}} -\mathbf{1}^\top \cdot \log\left(\mathbf{1} - \hat{\mathbf{y}}_{h,w}\right), \tag{14}$$

where $\mathbf{1}$ is a vector of all 1s. Note that Eq. (14) only penalizes wrong prediction of known relations between drug-pairs. In other words, for triplets that are not observed in $\mathcal{E}_{\mathrm{DDI}}$, we regard them as unknown.

During inference, given a new drug-pair $(h', t')$ where $h', t' \in \mathcal{V}$, we directly use KnowDDI with optimized $\boldsymbol{\theta}_g, \boldsymbol{\theta}_k$ to obtain the class prediction vector $\hat{\mathbf{y}}_{h',t'}$. For multiclass prediction, the class is predicted as the relation which obtains the highest possibility in $\hat{\mathbf{y}}_{h',t'}$. As for multilabel prediction, the complete $\hat{\mathbf{y}}_{h',t'}$ is returned. Please refer to Algorithm ?? in Supplementary Note 1 for details.

**Identifying explaining paths.** To explain the predicted DDI for $(h, t)$, we take out the explaining paths from $\mathcal{S}_{h,t}$. In particular, an explaining path

$$\mathbf{p}_{h,t} = h \xrightarrow{\mathbf{A}_{h,t}(h,v_1,r_1)} v_1 \xrightarrow{\mathbf{A}_{h,t}(v_1,v_2,r_2)} v_2 \tag{15}$$

$$\cdots \xrightarrow{\mathbf{A}_{h,t}(v_{P-1},t,r_P)} t, \tag{16}$$

is a sequence of nodes, where node $v_i$ and node $v_{i+1}$ are connected by relation $r_{i+1} \in \mathcal{R}$ with a connection strength indicated by $\mathbf{A}_{h,t}(v_i, v_j, r_j)$. We then obtain the average connection strength of $\mathbf{p}_{h,t}$ by averaging over $\mathbf{A}_{h,t}(v_i, v_j, r_j)$ of consecutive pairs of nodes in $\mathbf{p}_{h,t}$. This average connection strength reflects the ability of the explaining path to interpret the prediction result from the perspective of KnowDDI.

### Training details

In KnowDDI, we use a two-layer GraphSAGE[25] to obtain the generic node embedding $\mathbf{e}_v^{(l)}$ whose dimension is set as 32. For drug-flow subgraph extraction, we extract 2-hop neighborhood and then extract relational paths with length at most 4 pointing from $h$ to $t$. The dimension of edge embedding $\mathbf{h}_r$ in Eq. (6) is set as 32. We alternate between estimating connection strengths and refining node embeddings for 3 times ($T$ in Algorithm ??). We select $\gamma$ in Eq. (7) from [0.05, 0.2] and $\alpha$ in Eq. (10) from [0.3, 0.7]. We train the model for a maximum number of 50 epochs using Adam[28] with learning rate $5 * 10^{-3}$ and weight decay rate $10^{-5}$. We early stop training if the validation loss does not decrease for 10 consecutive epochs. We set dropout rate as 0.2 and batch size as 256. All results are averaged over five runs and are obtained on a 32GB NVIDIA Tesla V100 GPU. A summary of hyperparameters used by KnowDDI is provided in Supplementary Table 2. Their sensitivity analysis results are shown in Supplementary Fig. 2 and discussed in Supplementary Note 3.

### Reporting summary

Further information on research design is available in the Nature Portfolio Reporting Summary linked to this article.

## Results

### Data

In this study, we perform experiments on two publicly available benchmark DDI datasets: (i) Drugbank[13] is a multiclass DDI prediction dataset consisting of 86 types of pharmacological relations occurred between drugs; and (ii) TWOSIDES[29] is a multilabel DDI prediction dataset recording multiple DDI side effects between drugs. We adopt Hetionet[17,18], which is a benchmark biomedical KG for various tasks within drug discovery, as the external KG in this paper. Other recent developed biomedical KGs such as ogbl-biokg[30], OpenBioLink[31], and PharmKG[19] can also be used.

**Data preprocessing.** We preprocess the two benchmark DDI datasets DrugBank[13,32] and TWOSIDES[29,33] following the same procedure adopted by SumGNN[23]. In DrugBank, relations are skewed. Each drug-pair is filtered to have one relation only[23]. In TWOSIDES, 200 commonly occurring relations are selected. In particular, relations are ranked by decreasing number of associating fact triplets, and the 200 relations ranked between 600 to 800 are kept such that each relation is associated with at least 900 fact triplets[10]. Thus, relations in TWOSIDES are associated with comparable number of fact triplets. We formulate the benchmark DDI datasets as DDI graphs separately, whose statistics are summarized in Table 1. The fact triplets in DDI datasets are split into training, validation, and testing sets with a ratio of 7:1:2 following SumGNN[23] for fair comparison. We remove from external KG the drug-drug edges contained in DDI graph to avoid information leakage, then merge the resultant external KG and DDI graph into a large combined network. Eventually, the DDI graph of DrugBank is merged with a graph of 33765 nodes and 1690693 edges extracted from Hetionet, and the DDI graph of TWOSIDES is merged with a graph of 28132 nodes and 1666632 edges extracted from Hetionet, respectively. During training, the drug-drug edges in validation and testing sets are unseen. After tuning hyperparameters on fact triplets in validation set, the model performance is evaluated on fact triplets in testing set.

**Evaluation metric.** We evaluate the multiclass DDI prediction performance by three metrics: (i) Macro-averaged F1 which is averaged over class-wise F1 scores, (ii) Accuracy (ACC) which is the micro-averaged F1 score calculated using all testing fact triplets, and (iii) Cohen's $\kappa$ which measures the inter-annotator agreement. As for multilabel DDI prediction, we report the results averaged over all relation types. The performance is evaluated by (i) AUROC which is the average area under the receiver operating characteristics (ROC) curve, (ii) AUPRC which is the average area under the precision-recall (PR) curve, and (iii) AP@50 which is the average precision at 50.

### Comparison with the state-of-the-art

We consider multi-typed DDI prediction problem where interactions between drugs can have multiple relation types. For example, a drug-pair (drug A, drug B) can have two relation types "the matabolism of drug A can be decreased when combined with drug B" and "the therapeutic efficacy of drug A can be increased when combined with drug B". In particular, we compare the proposed KnowDDI with the following models:

- Traditional two-stage methods (w/o external KG). (i) KG embedding-based methods use shallow linear models to encode drug entities and their associated relations into low-dimensional embeddings, then feed the drug embeddings into a separately learned classifier for DDI prediction. Exemplar methods are TransE[34], KG-DDI[35,36], and MSTE[15,37]. (ii) Network embedding-based methods which use neural networks to encode structural information into node embeddings of drugs, and predict the relation types by a linear layer. Exemplar methods are DeepWalk[14,38], node2vec[39,40], and LINE[41,42].
- GNN-based methods (w/o external KG) formulate the existing DDI fact triplets into the form of a graph where each node corresponds to a drug and each edge between two drugs represents one relation type, then solve the resultant link prediction problem on the DDI graph using GNNs[43], including GAT[44,45], Decagon[10,46], and SkipGNN[11,47].
- GNN-based methods (w/ external KG) leverage an external KG to provide rich organized biomedical knowledge, and aggregate the messages from neighboring nodes of drugs by GNNs. Existing methods include GraIL[22,48], KGNN[21,49], DDKG[50,51], SumGNN[23,52], and LaGAT[24,53].

**Table 1 | Performance (%) is evaluated on two benchmark DDI datasets DrugBank and TWOSIDES**

| | Dataset | DrugBank (multiclass) | | | TWOSIDES (multilabel) | | | |
|---|---|---|---|---|---|---|---|---|
| | Statistics | $\|\mathcal{V}\| = 1710, \|\mathcal{R}\| = 86, \|\mathcal{E}\| = 192284$ | | | $\|\mathcal{V}\| = 604, \|\mathcal{R}\| = 200, \|\mathcal{E}\| = 41270$ | | | Avg. |
| | Metric | F1 | ACC | Cohen's $\kappa$ | AUROC | AUPRC | AP@50 | Rank |
| | TransE | 18.32 ± 0.16 | 64.60 ± 0.11 | 57.19 ± 1.22 | 77.53 ± 0.02 | 70.16 ± 0.02 | 77.54 ± 0.03 | 15 |
| Traditional | KG-DDI | 37.21 ± 0.36 | 82.74 ± 0.10 | 78.71 ± 0.33 | 90.64 ± 0.09 | 87.99 ± 0.11 | 83.50 ± 0.06 | 10 |
| Two-stage | MSTE | 53.96 ± 0.07 | 77.76 ± 0.11 | 73.35 ± 0.05 | 89.40 ± 0.04 | 83.06 ± 0.06 | 79.38 ± 0.03 | 12 |
| (w/o external | node2vec | 50.00 ± 1.67 | 62.39 ± 0.99 | 56.27 ± 0.89 | 90.62 ± 0.43 | 89.42 ± 0.47 | 82.21 ± 0.54 | 12 |
| KG) | DeepWalk | 49.17 ± 1.38 | 62.90 ± 0.37 | 56.77 ± 0.44 | 91.77 ± 0.26 | 90.56 ± 0.23 | 84.13 ± 0.35 | 9 |
| | LINE | 48.89 ± 1.38 | 59.87 ± 0.92 | 52.15 ± 1.51 | 88.63 ± 0.20 | 87.02 ± 0.22 | 80.80 ± 0.20 | 14 |
| GNN-based | GAT | 35.05 ± 0.41 | 78.02 ± 0.14 | 74.68 ± 0.21 | 91.22 ± 0.13 | 89.79 ± 0.10 | 83.05 ± 0.18 | 11 |
| (w/o external | Decagon | 56.24 ± 0.27 | 86.97 ± 0.31 | 86.12 ± 0.09 | 91.83 ± 0.14 | 90.79 ± 0.18 | 82.49 ± 0.36 | 7 |
| KG) | SkipGNN | 62.36 ± 0.96 | 88.04 ± 0.66 | 85.71 ± 0.81 | 92.31 ± 0.15 | 90.84 ± 0.03 | 84.23 ± 0.19 | 6 |
| | GraIL | 75.92 ± 0.69 | 89.63 ± 0.39 | 87.63 ± 0.47 | 93.73 ± 0.10 | 92.26 ± 0.07 | 86.89 ± 0.11 | 3 |
| GNN-based | KGNN | 74.08 ± 0.92 | 88.30 ± 0.08 | 86.09 ± 0.10 | 92.93 ± 0.10 | 90.11 ± 0.14 | 87.43 ± 0.09 | 5 |
| (w/ external | DDKG | 75.84 ± 0.22 | 88.70 ± 0.39 | 87.53 ± 0.21 | 93.15 ± 0.18 | 91.09 ± 0.39 | 87.50 ± 0.43 | 4 |
| KG) | SumGNN | <u>86.88 ± 0.63</u> | <u>91.86 ± 0.23</u> | <u>90.34 ± 0.28</u> | <u>94.61 ± 0.06</u> | <u>93.13 ± 0.15</u> | <u>88.38 ± 0.07</u> | 2 |
| | LaGAT | 83.69 ± 0.74 | 88.86 ± 0.12 | 87.33 ± 0.14 | 88.72 ± 0.22 | 84.03 ± 0.43 | 82.46 ± 0.41 | 8 |
| | KnowDDI | **91.53 ± 0.24** | **93.17 ± 0.09** | **91.89 ± 0.11** | **95.43 ± 0.02** | **94.14 ± 0.03** | **89.54 ± 0.03** | 1 |

Average (Avg.) rank of each method is reported in the last column, which is averaged over the six columns of performance. The best and comparable results (according to the pairwise t-test with 95% confidence) are highlighted in bold, and the second-best results are underlined. $|\cdot|$ counts the number of elements in a set. $\mathcal{V}$ is the set of drug nodes, $\mathcal{E}$ is the set of fact triplets, and $\mathcal{R}$ is the set of relation types.

We implement the baselines using public codes of the respective authors, except TransE[34] which is implemented by us.

**Overall performance.** Table 1 shows the results obtained on two benchmark DDI datasets. Overall, we can see that GNN-based methods (w/ external KG) generally perform the best and traditional two-stage methods generally perform the worst. Comparing with traditional two-stage methods, GNN-based methods (w/o external KG) can better propagate information among connected nodes (i.e., drugs) by modeling the fact triplets integrally as a graph and jointly learning all model parameters w.r.t. the objective in an end-to-end manner. However, due to the lack of DDI fact triplets and the over-parameterization of GNN, they may not consistently be better than traditional two-stage methods. This can be supported by the observation that DeepWalk and KG-DDI perform better than the deep GAT.

Next, GNN-based methods ((w/ external KG)) leverage rich biomedical knowledge to alleviate the data scarcity problem. Among these methods, KGNN performs the worst. In contrast to pure GNN, KGNN uniformly samples $N$ nodes as neighbors of each node during message passing to reduce computational overhead. DDKG improves KGNN by assigning attention weights to the $N$ nodes during message passing, where the attention weights are obtained by calculating the similarity between initial node embeddings constructed from SMILES. In the end, each drug obtains its representation without considering which drug to interact in KGNN and DDKG. While GraIL, SumGNN, LaGAT and our KnowDDI merge the DDI graph with an external KG as a large combined network, then learn to encode more local semantic information from the combined network by extracting subgraphs w.r.t. drug-pairs. A drug can be represented differently in different subgraphs. Thus, these methods can obtain drug-pair-aware representations that can be beneficial to predict DDI types. In particular, GraIL directly propagates messages on the extracted subgraphs, LaGAT aggregates messages with attention weights calculated using node embeddings, and SumGNN only prunes edges based on node features that are randomly initialized or pretrained on other tasks and then fixes the subgraphs. They all cannot adaptively adjust the structure of subgraphs during learning.

Finally, KnowDDI learns to remove irrelevant edges and add new edge of type "resemble" based on learned node embeddings. Upon the purified subgraph (i.e., knowledge subgraph) of target drug-pair, KnowDDI transforms generic node embeddings to be more predictive of DDI types. The performance gain of KnowDDI over existing methods validates its effectiveness.

**Relation-wise performance.** Next, we take a closer look at the performance gain w.r.t. different relations grouped by frequency. Figure 2a shows the relation-wise F1 score (%) grouped into bins according to the number of fact triplets associated with the relation. We compare the proposed KnowDDI with SkipGNN which performs the best among GNN-based methods (w/o external KG), and SumGNN which obtains the second-best among GNN-based methods (w/ external KG). In addition, we compare with KnowDDI (w/o resemble), a variant of KnowDDI which does not add new edges of type "resemble" between nodes with highly similar node embeddings.

As shown, by comparing the performance of SkipGNN and the other methods, we can see that external KG plays an important role. In general, KnowDDI and KnowDDI (w/o resemble) consistently obtain better performance than SumGNN, and the performance gap is larger on relations with fewer known fact triplets. This shows that enriched drug representations and adjusted subgraphs can be helpful to compensate for the lack of known DDI fact triplets. KnowDDI performs the best, which further shows the contribution of learning to propagate resembling relationships between highly similar nodes. Additionally, Supplementary Fig. 3 shows statistics of relation frequency and Supplementary Fig. 4 shows relation-wise performance improvement of KnowDDI over SumGNN on DrugBank, TWO-SIDES and a larger version of TWOSIDES with more relations. We also examine the performance of KnowDDI and the second-best method SumGNN obtained on some important and commonly studied adverse drug reactions (ADRs) in Supplementary Table 3. Results consistently show that KnowDDI obtains better performance. An extended discussion is provided in Supplementary Note 4.

**Compensating unknown DDIs.** Recall that the lack of DDI fact triplets is compensated by both enriched drug representations and propagated drug similarities in KnowDDI. Here, we pay a closer look to the effectiveness of these two designs. To achieve this goal, we examine the performance of proposed KnowDDI and KnowDDI (w/o resemble) with different amount of fact triplets introduced from external KG to the combined network. We compare them with SumGNN which obtains the second-best among GNN-based methods (w/ external KG), and take SkipGNN from GNN-based methods (w/o external KG) as a reference. Figure 2b plots the performance changes w.r.t varying portion (%) of fact triplets sampled from the external KG. First, SumGNN, KnowDDI and KnowDDI (w/o resemble) all perform worse given fewer fact triplets from the external KG. This is because a sparser external KG means less information introduced into DDI datasets, which reduces the

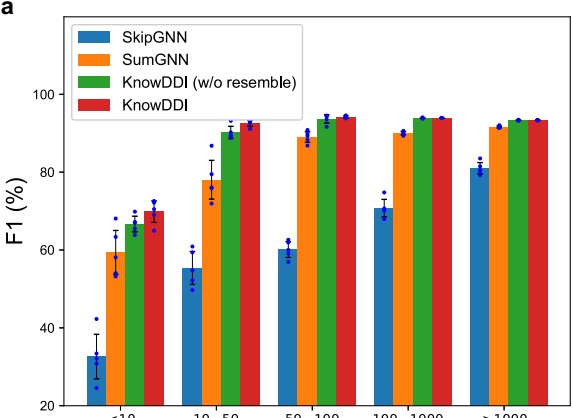

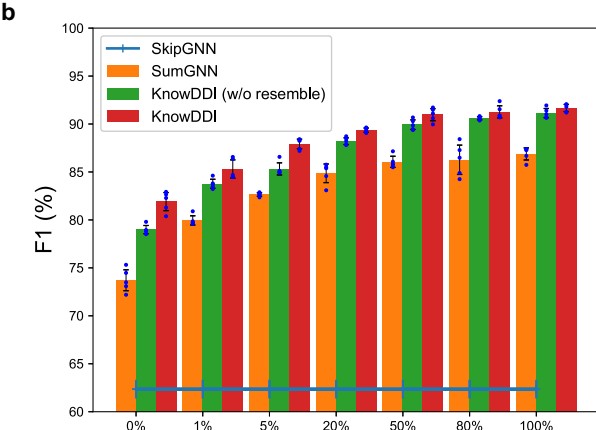

**Fig. 2 | A closer examination and comparison of KnowDDI with SumGNN, SkipGNN and KnowDDI (w/o resemble). a** F1 (%) obtained for relations with different number of DDI fact triplets on DrugBank. **b** F1 (%) obtained with different portion (%) of fact triplets sampled from the external KG on DrugBank. SkipGNN,

which does not use external KG, is plotted just for reference. This bar plot illustrates the test performance (%), with each bar's height representing the mean result and the error bars indicating the standard deviation, both derived from five independent runs ($n = 5$).

information gap between GNN-based methods w/ or w/o external KG. Then, KnowDDI (w/o resemble) consistently outperforms the other two methods as it keeps information that is more relevant to predicting DDI for the drug-pair at hand both during subgraph construction and learning. Besides, KnowDDI is the best, as it further learns drug similarities by propagating resembling relationships between drugs with highly similar representations. As a result, KnowDDI suffers the least from a sparser KG. However, the performance gap between KnowDDI (w/o resemble) and KnowDDI gets larger with fewer triples. This means removing irrelevant edges and propagating drug similarities have a larger influence on compensatingq for the lack of DDIs when drug representations are less enriched. Finally, let us pay special attention to the case of given 0% triples, which means external KG is not used. Still, we can observe that both KnowDDI (w/o resemble) and SumGNN still outperform SkipGNN. This can be attributed to the different subgraph extraction strategies adopted in these three methods, which will be carefully examined in Section 6.

### Effectiveness of knowledge subgraph
Here, we pay a closer look at knowledge subgraph $\mathcal{S}_{h,t}$ designed in KnowDDI, and compare it with other choices of subgraphs in terms of performance and interpretability.

**Subgraph extraction strategy.** To empirically validate the effectiveness of the proposed knowledge subgraph, we consider the following subgraphs: (i) Random subgraph consists of a fixed-size set of nodes uniformly sampled from the neighborhoods of $h$ and $t$ in $\mathcal{G}$, which is adopted in KGNN[21] to reduce the computation overhead; (ii) Enclosing subgraph is the interaction of $K$-hop neighborhoods of $h$ and $t$ in $\mathcal{G}$, which is adopted in GraIL[22], and SumGNN[23]; (iii) Drug-flow subgraph $\bar{\mathcal{S}}_{h,t}$ consists of relational paths pointing from $h$ to $t$ in $\mathcal{G}$ with length at least $P$; and (iv) Knowledge subgraph $\mathcal{S}_{h,t}$ consists of explaining paths from $h$ to $t$, which is obtained by iteratively refining the graph structure and node embeddings of $\bar{\mathcal{S}}_{h,t}$. Apart from them, we further compare with the knowledge subgraphs obtained by KnowDDI (w/o resemble), and denote the results as knowledge subgraph (w/o resemble).

Figure 3a shows the results obtained by KnowDDI on DrugBank with different subgraphs and different percentages of fact triplets from external KG. Enlarged enclosing subgraphs are provided in Supplementary Fig. 5. As shown, leveraging subgraphs consistently leads to performance gain, regardless of the subgraph type. This shows the necessity of modeling local contexts of target drug-pairs. Among these subgraphs, learning with knowledge subgraph obtains the best performance. As random subgraph consists of uniformly sampled nodes without considering the node importance, the selected nodes may not contribute to recognize the relationships between head and tail drugs. Enclosing subgraph keeps the local neighborhood of head and tail drugs intact, thus it does not lose information. However, directly learning on these subgraphs may lead to bad performance, if irrelevant edges exist. In contrast, drug-flow subgraph focuses on relational paths pointing from head drug to tail drug, and knowledge subgraph further only keeps explaining paths. They all remove irrelevant nodes which do not appear in any paths. Besides, by comparing the performance of drug-flow subgraph and knowledge subgraph under different percentages of fact triplets, we can see that both enriched drug representations and propagated drug similarities contribute to the performance improvements. However, the performance gain is larger when fewer fact triplets are used. This means removing irrelevant edges and propagating drug similarities play a stronger influence on compensating for the lack of DDIs when drug representations are less enriched. In summary, learning knowledge subgraph is effective.

**Interpretability.** As discussed, being able to understand the DDI between drug-pairs helps drug discovery. Here, we show that KnowDDI can explain why two drugs associate with each other by leveraging explaining paths in knowledge subgraphs $\mathcal{S}_{h,t}$. Figure 3 shows the subgraphs of four drug-pairs. Random subgraphs are not plotted, as they naturally loses semantic information. As can be observed, drug-flow subgraphs contain fewer nodes in comparison to the enclosing subgraphs. Particularly, as we take Hetionet as the external KG, where only drugs have incoming edges with drugs and the relation type is "Compound-resembles-Compound", drug-flow subgraphs only contain drugs. Knowledge subgraphs further adjust the graph structure. In particular, KnowDDI assigns a connection strength between each node-pair from both direction. It represents the importance of a given edge between two connected nodes in the drug-flow subgraph or similarity between two nodes whose interactions are unknown. Even if two nodes are connected in the $\bar{\mathcal{S}}_{h,t}$, KnowDDI can delete an existing edge if the estimated connection strength is too small, such as the edge pointing from 575 to 284 in Fig. 3d. Likewise, two originally disconnected nodes can be connected after learning, if the estimated connection strength is large. This reveals that KnowDDI thinks the connected drugs are highly similar and can contribute to explaining the DDI type between two drugs, such as the edge pointing from 121 to 622 in Fig. 3c. Supplementary Fig. 6 shows the node-pair whose connection strength is the largest on each of knowledge graph, including their molecular graphs and drug efficacy.

Further, Table 2 shows the explaining paths with the largest average connection strengths assigned by KnowDDI (see Section 6) for the four drug-pairs in Fig. 3. We use Hetionet KG and DrugBank database to help interpret these explaining paths. As can be seen, these explaining paths indeed discover reasonable explanations. Moreover, note that in the second drug-pair, without the newly added edge of relation type "resemble" pointing from 121 to 622, the discovered explaining path no longer exists. This validates the necessity of learning with knowledge subgraphs.

### Embedding visualization
Finally, we wish to show that KnowDDI helps better shape the embeddings of drug-pairs and relations to be more predictive of DDI types between target drug-pairs. From the DDI dataset, we randomly sample ten DDI relations, then randomly sample fifty fact triplets per relation. Given a drug-pair $(h, t)$, it is represented as the concatenation of drug embeddings which correspond to the node embeddings of $h$ and $t$. For methods operating on subgraphs, the drug-pair embedding is obtained by concatenating drug embeddings with subgraph embedding which is obtained by mean pooling over node embeddings of all nodes in the subgraph[22,23]. In this way, local context of target drug-pair is leveraged to obtain better prediction results. We compare KnowDDI with SumGNN and KnowDDI (w/o resemble). In addition, we also show the drug-pair embeddings obtained by simply learning generic node embeddings without refining them on subgraphs. Figure 4 shows the t-SNE visualization[54] obtained on DrugBank. First, we can see from generic embeddings in Fig. 4b, to KnowDDI (w/o resemble) in Fig. 4c, to KnowDDI in Fig. 4d, drug-pair with the same relation type are getting closer while drug-pair with different relation types are moving farther apart. Also, as can be seen, the clusters are more obvious in KnowDDI (Fig. 4c, d) than that of SumGNN (Fig. 4a). This means that learning knowledge subgraphs is beneficial to obtain more distinctive drug-pair embeddings.

## Discussion
In this study, we are motivated to develop an effective solution to accurately construct a DDI predictor from the rare DDI fact triplets. The proposed KnowDDI achieves the goal by taking advantage of rich knowledge in biomedicine and healthcare and the plasticity of deep learning approaches. In KnowDDI, the enriched drug representations and propagated drug similarities together implicitly compensate for the lack of known DDIs. We first combine the provided DDI graph and an external KG into a combined network, and manage to encode the rich knowledge recorded in KG into the generic node representations. Then, we extract a drug-flow subgraph for each drug-pair from the combined network, and learn a knowledge subgraph from generic representations and the drug-flow subgraph. During learning, the knowledge subgraph is

**a**

| F1 | w/o Subgraph | Random Subgraph | Enclosing Subgraph | Drug-flow Subgraph | Knowledge Subgraph |
|---|---|---|---|---|---|
| 100% | 86.74 ± 0.48 | 89.49 ± 0.59 | 90.03 ± 0.28 | 90.63 ± 0.20 | 91.53 ± 0.24 |
| 5% | 80.77 ± 1.19 | 84.82 ± 0.92 | 85.06 ± 0.50 | 85.32 ± 0.63 | 87.92 ± 0.50 |
| 0% | 72.71 ± 0.92 | 75.88 ± 1.16 | 77.74 ± 0.25 | 78.99 ± 0.43 | 81.91 ± 0.94 |

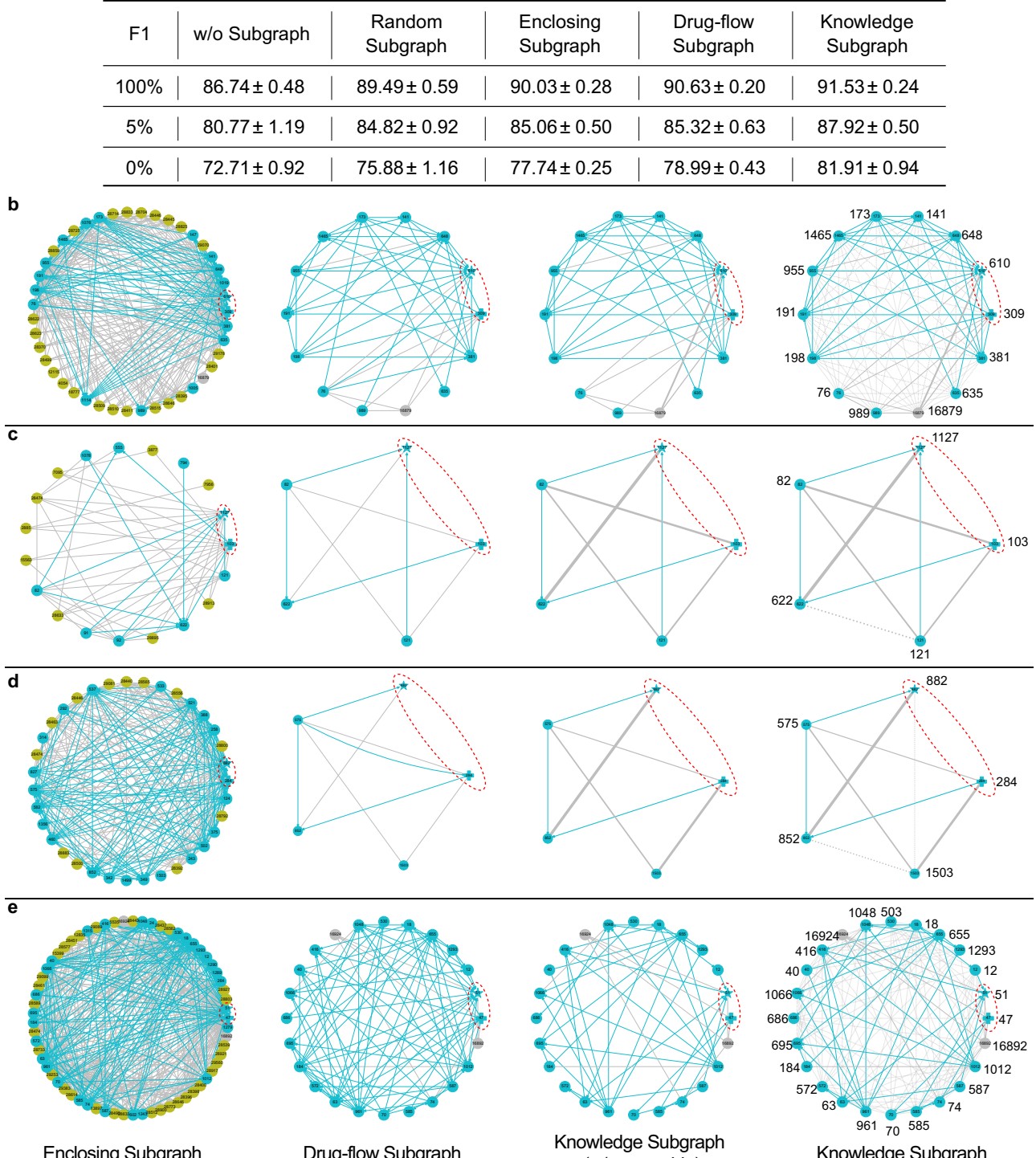

Enclosing Subgraph Drug-flow Subgraph Knowledge Subgraph (w/o resemble) Knowledge Subgraph

**Fig. 3 | Comparing different subgraphs extraction strategy. a** F1 (%) obtained with different portion (%) of fact triplets sampled from the external KG on DrugBank. **b–e** four exemplar drug-pairs $(h, t)$s in DrugBank and their enclosing subgraphs (first column), drug-flow subgraphs (second column), knowledge subgraphs without propagating resembling relationships (third column), and knowledge subgraphs (fourth column). Drugs from DDI graph are colored in blue, drugs from external KG are colored in gray, while the other entities are colored in green. In particular, we draw a red dashed circle encompassing the target drug-pair, and mark the head drug $h$ by filled plus and the tail drug $t$ by star. We mark edges from DDI graph and external KG in solid blue lines and solid gray lines respectively. We specially use undirected edges to represent relation type"resemble" as the semantics of resemble are not directive, and plot new edges of relation type"resemble" estimated by KnowDDI with dotted gray lines. The thickness of an edge is used to reveal the magnitude of the connection strength learned by KnowDDI.

optimized, where irrelevant edges are removed and new edges are added if two disconnected nodes have highly similar node representations. Finally, the representations of drugs are transformed to be more predictive of the DDI types between the target drug-pair while knowledge subgraph contains explaining paths to interpret the prediction result. The performance gap between KnowDDI and other approaches gets larger for relation types given a smaller number of known DDI fact triplets, which validates the effectiveness of KnowDDI.

**Table 2 | Explaining paths with the largest average connection strengths assigned by KnowDDI for four drug-pairs in DrugBank**

| | |
|---|---|
| Drug-Pair | (309,610) |
| DDI Type | The metabolism of Atomoxetine (610) can be decreased when combined with Reboxetine (309). |
| Explaining Path | $309 \xrightarrow{resembles} 16879 \xrightarrow{resembles} 610$ |
| Explanation | Reboxetine (309) resembles Diphenylpyraline (16879), while Diphenylpyraline (16879) resembles Atomoxetine (610). We can deduce that Reboxetine (309) resembles Atomoxetine (610). When taking two similar drugs, the body may absorb less of the Atomoxetine (610). |
| Drug-Pair | (103,1127) |
| DDI Type | The serum concentration of Betamethasone (103) can be increased if combined with Estriol (1127). |
| Explaining Path | $103 \xrightarrow{resembles} 121 \xrightarrow{resembles(newedge)} 622 \xrightarrow{resembles} 1127$ |
| Explanation | Betamethasone (103) bears resemblance to Desonide (121). Although the interaction between Desonide (121) and Mestranol (622) is not provided, our search within DrugBank reveals that combining Mestranol (622) with Budesonide can elevate the serum concentration of Budesonide. Furthermore, Budesonide is similar to Desonide (121), and Mestranol (622) is similar to Estriol (1127). Given the tendency for similar drugs to exhibit akin properties, it is plausible that the serum concentration of Betamethasone (103) could rise when used alongside Estriol (1127). |
| Drug-Pair | (284,882) |
| DDI Type | The therapeutic efficacy of Sulfadiazine (284) can be increased when used in combination with Gatifloxacin (882). |
| Explaining Path | $284 \xrightarrow{therapeuticefficacyincreased} 852 \xrightarrow{resembles} 882$ |
| Explanation | The combined use of Sulfadiazine (282) and Pefloxacin (852) can improve the efficacy. Meanwhile, Pefloxacin (852) resembles Gatifloxacin (882). It can be deduced that the therapeutic efficacy of Sulfadiazine (282) can be increased when used in combination with Gatifloxacin (882). |
| Drug-Pair | (47,51) |
| DDI Type | The risk or severity of adverse effects can be increased when Atropine (47) is combined with Scopolamine (51). |
| Explaining Path | $47 \xrightarrow{resembles} 16892 \xrightarrow{resembles} 51$ |
| Explanation | Atropine (47) resembles Homatropine methylbromide (16892), while Homatropine methylbromide (16892) resembles Scopolamine (51). We can deduce that Atropine (47) resembles Scopolamine (51). The similarities between the two drugs can also be seen through the chemical structures of the drugs. As the two drugs are similar in structure, the effects of using both drugs at the same time should be similar to the side effects of overdose of either drug, which can cause serious side effects. Based on DrugBank, Scopolamine (51) overdose may manifest as lethargy, somnolence, coma, confusion, agitation, hallucinations, convulsion, visual disturbance, dry flushed skin, dry mouth, decreased bowel sounds, urinary retention, tachycardia, hypertension and supraventricular arrhythmias, while Atropine (47) overdose may cause palpitation, dilated pupils, difficulty swallowing, hot dry skin, thirst, dizziness, restlessness, tremor, fatigue and ataxia. |

Possible explanations are discovered from Hetionet and DrugBank.

Due to the popularity of GNN for learning from graphs, existing works have applied it to solve link prediction problem. Early works, like GAT[44] and R-GCN[55], usually obtain the representation for each node by running message passing on the whole graph, then feed the representation of two target nodes into a predictor to estimate the existence of a link between two target nodes. Decagon[10], SkipGNN[11], KGNN[21] and DDKG[50] compared in Table 1 also follow this routine. In particular, KGNN uniformly samples a fixed number of nodes as neighbors of each node during message passing to reduce the computation overhead. DDKG improves the message passing part of KGNN by assigning attention weights to the uniformly sampled neighboring nodes, where the attention weights are obtained by calculating the similarities between initial node embeddings constructed from SMILES. These works treat all nodes equally and ignore pair-wise information when propagating messages. Each drug will get a representation without considering which drug to interact. Thus, the performance of these methods can be worse than classical hand-designed heuristics, which count common neighbors or connected paths between a node-pair[56]. As a result, recent work GraIL[22] proposes a pipeline to learn with subgraphs, i.e., first extracting a subgraph containing the two target nodes, then obtaining the node representation from the subgraph, finally estimating the link between two target nodes using the node-pair representation which consists of the node embeddings of two target nodes and the subgraph embedding. KnowDDI, SumGNN[23] and LaGAT[24] compared in Table 1 follow this pipeline. They adopt different strategies to learn with subgraphs. In particular, LaGAT extracts a subgraph consisting of a fixed number of nodes around the head and tail drugs, updates node embeddings by aggregating neighboring nodes based on attention weights calculated using node embeddings, and leaves the subgraph unchanged. While SumGNN extracts the enclosing subgraph of each drug-pair, then prunes edges based on the node features. By encoding local context within subgraphs, these methods obtain node-pair-aware representations, i.e., a drug can be represented different depending on

which drug to interact. Our KnowDDI also learns with subgraphs, while two major design differences makes it obtain better and more interpretable results. The first difference is that KnowDDI learns generic node embeddings on the combined network to enrich the drug representations, then transforms them on knowledge subgraphs to incorporate with the local context of drug-pairs. The second difference is the adjustment of subgraphs where existing edges can be dropped if their estimated importance are low, and new edges of type "resemble" can be added between disconnected nodes if their node embeddings are highly similar. This allows KnowDDI to capture explaining paths pointing from head drug to tail drug. While SumGNN directly learns drug-pair-aware representations from the extracted subgraphs. With the two differences, KnowDDI achieves the balance of generic information and drug-pair-aware local context during learning.

The architecture of KnowDDI can be further improved. For instance, pretraining GNN from other large datasets which may provide better initialized parameters and therefore reduce the training time. Besides, we do not use any molecular features of drugs in order to test the ability of KnowDDI learning solely from the combination of external KG and DDI fact triplets. Taking these predefined node features may improve the predictive performance of KnowDDI in the future. Although we implement KnowDDI to handle DDI prediction in this paper, KnowDDI is a general approach which can be applied to other relevant applications, to help detect possible protein-protein interactions, drug-target interactions, and disease-gene interactions. Relevant practitioners can easily leverage the rich biomedical knowledge existing in large KGs to obtain good and explainable prediction results. We believe our open-source KnowDDI can act as an original algorithm and unique deep learning tool to promote the development of biomedicine and healthcare. For example, it can help detect possible interactions of new drugs, accelerating the speed of drug design. Given drug profiles of patients, KnowDDI can be used to identify possible adverse

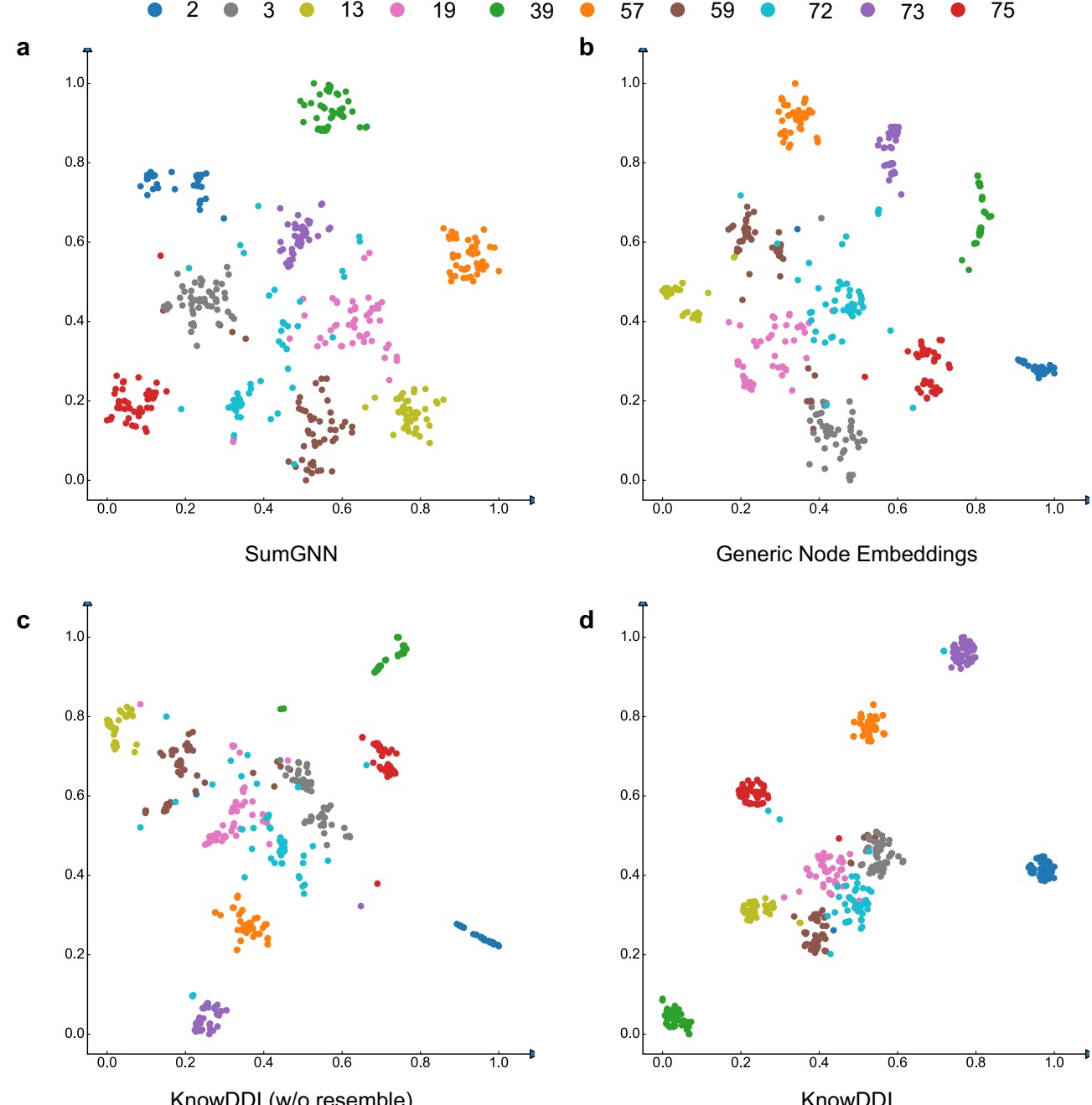

**Fig. 4 | T-SNE visualization of drug-pair embeddings, where the four graphs share the same relations ($n = 10$) and drug-pairs (for each relation, $n = 50$). a** T-SNE of drug-pair embeddings obtained by SumGNN. **b** T-SNE of drug-pair embeddings which are composed of concatenated generic node embeddings. **c** T-SNE of drug-pair embeddings obtained by KnowDDI (w/o resemble). **d** T-SNE of drug-pair embeddings obtained by KnowDDI.

reactions. These results have the potential to serve as a valuable resource for alerting clinicians and healthcare providers when devising management plans for polypharmacy, as well as for guiding the inclusion criteria of participants in clinical trials. Beyond biomedicine and healthcare, similar approaches can be developed to adaptively leverage domain-specific large KGs to help solve downstream applications in low-data regimes.

## Data availability

All data used in this study are available in supplementary data and public repositories. Source data underlying Figs. 2–4 can be found in Supplementary Data 1. For the benchmark DDI datasets, DrugBank dataset[13] can be downloaded from https://bitbucket.org/kaistsystemsbiology/deepddi/src/master/data/[32], and TWOSIDES dataset[29] can be downloaded from https://tatonettilab.org/resources/nsides/[33]. The external KG Hetionet[17] is obtained from https://het.io[18]. The processed data analyzed in this paper is available in GitHub repository at https://github.com/LARS-research/KnowDDI/tree/main/data[57].

## Code availability

The code implementing KnowDDI is deposited in public available GitHub repository at https://github.com/LARS-research/KnowDDI[58]. The version for this publication is provided in Zenodo with the identifier: https://doi.org/10.5281/zenodo.10285646[59].

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

## Acknowledgements
Q.Y. is supported by research fund of National Natural Science Foundation of China (No. 92270106), and Independent Research Plan of the Department of Electronic Engineering Department at Tsinghua University,

## Author contributions
For this manuscript, Y.W. contributes to the idea development, experiment design, paper presentation and writing; Z.Y. contributes to the code implementations, obtaining and analysis results; Q.Y. contributes to the idea development, result analysis, paper presentation and writing.

## Competing interests
The authors declare no competing interests.
