## [Peer Review File · Communications Medicine]

Reviewers' comments:

Reviewer #1 (Remarks to the Author):

This work presents a deep learning method, named KnowDDI, to predict DDIs from a given biomedical knowledge graph. KnowDDI is also an interpretable model and can be used to explain the prediction results for drug pairs through reasonable paths. It has been evaluated on two scenarios and it yields better performance than existing SOTA methods on several benchmark datasets. There are several major concerns regarding the quality of this work, and they are listed as below.

1. In the subsection of "Drug Pair-aware Subgraph Construction", the generated relational path starts with the drug h , but ends with $v_{\{P\}}$. Is drug t supposed to be the tail node.
2. In the subsection of "Knowledge Subgraph Generation", authors have reported that the triplet in the $\bar{S}_{\{h,t\}}$ is evaluated by formulation (3). Is this evaluation applied to any two nodes in the $\bar{S}_{\{h,t\}}$, even if there is no connection between u and v ?
 - 1) If so, the authors are supposed to clearly declare it in the manuscript. Moreover, how do you evaluate the relation r ?
 - 2) If not, how do you add new connections or those missing edges to the new graph $S_{\{h,t\}}$? The new adjacent matrix $A_{\{h,t\}}$ cannot contain missing edges according to the formulation (4) from my point of view, which is not in line with the statement of authors "this means there should be edges between u and v which are missing in $\bar{S}_{\{h,t\}}$ ". I don't think that KnowDDI can be able to capture missing edges.
3. Fig. 3b-d is not easy to understand. First, authors are supposed to claim that these three subfigures are presented for three samples, maybe you can draw horizontal lines instead of vertical lines to distinguish node pairs. The vertical lines are misleading. Second, the drug pair is supposed to be enlarged. Third, why are edges in the knowledge subgraph bidirectional? The importance of each edge is calculated based on the triple fact and the graph used in your work should be a directed graph.
4. In the section of "Embedding visualization", Fig. 4a is meaningless from my point of view. This example cannot show the advantage of KnowDDI nor its interpretability. The KGNN has the same ability or those spatial-based GNN methods can also learn different embeddings for drugs in two different triplets. Moreover, it has the same problem of Fig. 3. The authors are supposed to enlarge the drug nodes.
5. How about removing the Generic node embedding generation? Is it necessary?
6. What is the effect of different lengths of the path on the learned embedding?
7. Why do the authors integrate subgraph embedding into the calculation of $\hat{y}_{\{h,t\}}$? Please explain the rationality behind this integration.
8. Some recently published papers that also make use of knowledge graph for DDI prediction are missed for discussion in related work, such as DOI: 10.1093/bib/bbac140.

Reviewer #2 (Remarks to the Author):

The work proposed a knowledge graph method named KnowDDI to predict DDI interactions. It exploits large external KG to learn drug pair knowledge subgraphs from the KG and obtain node's embedding in KG. And it utilizes explaining paths generated from KG to interpret the prediction results for drug pairs.

However, this work should consider the following comments:

λ In the section of Abstract, the purpose of introducing KG is not to compensate for the lack of

known DDIs, but to obtain the rich neighborhood information of each entity in KG.

λ On line 101, it says “In addition, they are not developed for a particular downstream task such as DDI prediction considered in this paper, the mismatch of objectives can lead the deep learning model to learn irrelevant or distractive knowledge”. But both the methods of sum GNN [21] and KGNN [30] are constructed for the downstream task of DDI prediction. The method of KnowDDI in this manuscript is not the first one.

λ The section Dataset about data extract and description is missing in the manuscript.

λ With the representation learned from the previous sections, how to predict the interaction value between drug-drug pairs, including the multi-class DDI prediction and the multi-label DDI prediction? This is not illustrated in the manuscript.

λ In line 382, it uses the assumption of “similar drugs may have similar properties” to explain the interaction between Drug 103 and Drug 1127, which seems strained. Because it explains drug 103 is similar to drug 82, drug 82 is similar to drug 121, drug 121 is similar to drug Budesonide, drug 622 is similar to drug 1127, and there is no interaction between drug 121 and drug 622. Therefore, it is too strained to explain the occurrence reason of DDI by so many similarity hops, which oppositely reflects the difference between drugs in many cases. It is best to find literature and database to support their conclusion. In addition, why is it that only drug nodes are found in the interpretable path, and there are no other entity nodes of the KG, such as proteins and diseases? Adding other entity nodes can mine more meaningful interpretability of DDI.

Reviewer #3 (Remarks to the Author):

****Major Claim.**** To predict drug-drug interactions, the authors propose KnowDDI, a deep learning method that leverages large biomedical KGs for prediction and generates potential explaining paths. They demonstrate by experiments that KnowDDI outperforms existing works regarding prediction power, interpretability and robustness to sparse data.

****Novelty.**** As the authors claim, KnowDDI is the first deep method exploiting biomedical KGs for DDI prediction. They make dedicated designs to transform the noisy biomedical KGs into operatable knowledge graphs for the DDI prediction. The novelty could be enhanced with detailed discussion about KnowDDI's connections to previous GNN-based DDI methods with external KG.

****Convincingness.**** The empirical experiments are extensive and convincing. The methodology part is developed in a clear structure. However, as the transformation from external KGs to operatable graphs lies in the core of the whole method, related formulation and statement could be more detailed. Specifically, more introduction about the MLP adopted to estimate relevance scores of relations would facilitate the illustration of the refining process, along with its role in the entire method. Also, careful proofreading would be helpful. Several notations are used without clarification, e.g., $\$AGG(\cdot)\$$ in Eq.1, $\$COMB(\cdot)\$$ in Eq.2.

****Reproduction.**** I think KnowDDI is reproducible. It would be much easier with code and data made available.

Reviewer #4 (Remarks to the Author):

The paper proposes a deep learning method for DDI prediction with an external biomedical knowledge graph. The proposed method first extracts the relevant subgraph of a given drug-drug pair from the entire KG and then iteratively refines the subgraph and maintains the useful information for prediction. Experiments are conducted on two benchmark DDI datasets.

Major comments:

1. This paper is mainly based on existing work (SumGNN) with minimum modification. The dataset and experimental setting are exactly the same. The only difference between the proposed method and SumGNN is the missing edge imputation during the knowledge subgraph generation, which has been widely studied in knowledge graph completion.

- An ablation study of comparing the model w/o missing edge imputation (only keeping irrelevant edge pruning) and SumGNN is required to demonstrate the model performance.

- Even though, the results in Fig. 3a show that the performance of the model w/ knowledge subgraph is comparable to the model w/ drug pair-aware subgraph (i.e., <1% improvement).

2. In Table 1, the performance improvement over the best baseline in TWOSIDES is relatively small (i.e., ~1%). Given this limited performance gain, it's doubtful that the proposed method can robustly and consistently achieve good performance on other different datasets.

3. Missing baselines. Some related work in DDI prediction with biomedical KG is neither cited nor compared in the paper.

- Hong, Yue, et al. "LaGAT: link-aware graph attention network for drug-drug interaction prediction." *Bioinformatics* 38.24 (2022): 5406-5412.

- Ren, Zhong-Hao, et al. "A biomedical knowledge graph-based method for drug-drug interactions prediction through combining local and global features with deep neural networks." *Briefings in Bioinformatics* 23.5 (2022).

4. The scores in Table 1 are not clinically meaningful - DDI prediction performance for each class (86 in DrugBank and 200 in TWOSIDES) is necessary to be included. More discussions of the model performance with respect to each class of DDI are expected, especially whether the model can achieve consistent performance in those relatively rare interactions. Also, model performance on some important and commonly studied adverse drug reactions (e.g., acute liver failure, acute myocardial infarction, acute renal failure and upper GI bleeding) needs to be discussed.

5. The code and data are not provided for review.

Rebuttal letter for “Adaptive usage of large biomedical knowledge graph enables accurate and interpretable drug-drug interaction prediction”

In the rebuttal letter, we provide point-by-point response to reviewers. The reviewers’ comments are in blue, and our replies are in black.

General Response

Q1. Discussion that puts this study in the context of existing literature

Reply. In the revised manuscript, we add the following discussion with related works in the second paragraph of Discussion section to discuss the connections and differences between KnowDDI and existing works.

“ Due to the popularity of GNN for learning from graphs, existing works have applied it to solve link prediction problem. Early works, like GAT [32] and R-GCN [35], usually obtain the representation for each node by running message passing on the whole graph, then feed the representation of two target nodes into a predictor to estimate the existence of a link between the two target nodes. Decagon [10], SkipGNN [11], KGNN [20] and DDKG [33] compared in Table 1 also follow this routine. In particular, KGNN uniformly samples a fixed number of nodes as neighbors of each node during message passing to reduce the computation overhead. DDKG improves the message passing part of KGNN by assigning attention weights to the uniformly sampled neighboring nodes, where the attention weights are obtained by calculating the similarity between initial node embeddings constructed from SMILES. These works treat all nodes equally and ignore pair-wise information when propagating messages. Each drug will get a unique representation, without considering which drug to interact. Thus, the performance of these methods can be worse than classical hand-designed heuristics, which count common neighbors or connected paths between a node-pair [36]. As a result, recent work GraIL [21] proposes a pipeline to learn with subgraphs, i.e., first extracting a subgraph containing the two target nodes, then obtaining the node representation from the subgraph, finally estimating the link between two target nodes using the node-pair representation which consists of the node embeddings of two target nodes and the subgraph embedding. KnowDDI, SumGNN [22] and LaGAT [23] compared in Table 1 follow this pipeline. They adopt different strategies to learn with subgraphs. In particular, LaGAT extracts a subgraph consisting of a fixed number of nodes around the head and tail drugs, updates node embeddings by aggregating neighboring nodes based on attention weights calculated using node embeddings, and leaves the subgraph unchanged. While SumGNN extracts the enclosing subgraphs of the drug-pair, then prunes edges based on the node features. By encoding local context within the subgraph, these methods obtain node-pair-aware representations, i.e., a drug can be represented different depending on which drug to interact. Our KnowDDI also learns with subgraphs, while two major design differences makes it obtain better and more interpretable performance. The first difference is that KnowDDI first learns generic node embeddings on the combined network which enriches the drug representation, then transforms them on knowledge subgraph to incorporate with the local context of drug-pair. The second difference is the adjustment of subgraph where existing edges can be dropped if their estimated importance are low, and new edges of type “resemble” can be added between disconnected nodes if their node embeddings are highly similar. This allows KnowDDI to capture explaining paths pointing from head drug to tail drug. While SumGNN directly learns drug-pair-aware representations from the extracted subgraph. With the two differences, KnowDDI achieves the balance of generic information and drug-pair-aware local context during learning. ”

We also provide a summary of characteristics comparing KnowDDI with existing works on solving DDI prediction tasks in Supplementary Table 1 (reproduced as Table R1).

Table R1: Comparing the proposed KnowDDI with existing works.

Method	External KG	Drug-pair-aware Representation	Adding New Edges	Removing Irrelevant Edges	Interpretable Paths
GAT [32]	✗	✗	✗	✗	✗
Decagon [10]	✗	✗	✗	✗	✗
SkipGNN [11]	✗	✗	✗	✗	✗
GraIL [21]	✓	✗	✗	✗	✗
KGNN [20]	✓	✗	✗	✗	✗
DDKG [33]	✓	✗	✗	✗	✗
SumGNN [22]	✓	✓	✗	✓	✓
LaGAT [23]	✓	✓	✗	✗	✓
KnowDDI	✓	✓	✓	✓	✓

Q2. Detailed methods section that describe the dataset and extraction methods.

Reply. In this paper, we use two benchmark DDI datasets DrugBank and TWOSIDES and following the procedure adopted by SumGNN to process the data. In the revised manuscript, we p

Q3. Full code and dataset placed in appropriate public depository and made available for reviewers and readers.

Reply. We add Data availability section and Code availability section in the revised manuscript.

*“ **Data availability.** The data used in this study, including the two benchmark DDI datasets and external KG Hetionet, is publicly available. DrugBank dataset [13] can be downloaded from <https://bitbucket.org/kaistsystemsbiology/deepddi/src/master/data/>. TWOSIDES dataset [24] can be downloaded from <https://tatonettlab.org/resources/nsides/>. Hetionet is obtained from <https://het.io>. The processed data analyzed in this paper is available in GitHub repository at https://github.com/LARS-research/KnowDDI/raw_data. Combined networks which separately merges the two DDI graph with external KG are provided at <https://github.com/LARS-research/KnowDDI/data>.*

***Cod availability.** The code implementing KnowDDI is deposited in public available GitHub repository at <https://github.com/LARS-research/KnowDDI>. ”*

Q4. Demonstrate how this approach can be made accessible for clinical use.

Reply. In the revised manuscript, we add the following discussion about possible clinical use of KnowDDI in the last paragraph in Discussion section.

“ We believe our open-source KnowDDI can act as an original algorithm and unique deep learning tool to promote the development of biomedicine and healthcare. For example, it can help detect possible interactions of new drugs, accelerating the speed of drug design. Given drug profiles of patients, KnowDDI can be used to identify possible adverse reactions. These results have the potential to serve as a valuable resource for alerting clinicians and healthcare providers when devising management plans for polypharmacy, as well as for guiding the inclusion criteria of participants in clinical trials. ”

To Reviewer #1

Q1. In the subsection of “Drug Pair-aware Subgraph Construction”, the generated relational path starts with the drug h , but ends with v_P . Is drug t supposed to be the tail node.

Reply. Yes. The path starts from h and ends with t . We change the corresponding part as follows (page 13): $\bar{p}_{h,t} = h \xrightarrow{r_1} v_1 \xrightarrow{r_2} v_2 \cdots \xrightarrow{r_P} t$.

Q2. In the subsection of “Knowledge Subgraph Generation”, authors have reported that the triplet in the $\bar{S}_{h,t}$ is evaluated by formulation (3). Is this evaluation applied to any two nodes in the $\bar{S}_{h,t}$, even if there is no connection between u and v ?

- 1) If so, the authors are supposed to clearly declare it in the manuscript. Moreover, how do you evaluate the relation r ?
- 2) If not, how do you add new connections or those missing edges to the new graph $S_{h,t}$? The new adjacent matrix $A_{h,t}$ cannot contain missing edges according to the formulation (4) from my point of view, which is not in line with the statement of authors “this means there should be edges between u and v which are missing in $\bar{S}_{h,t}$ ”. I don’t think that KnowDDI can be able to capture missing edges.

Reply. We sincerely thank the reviewer for this question! Your comments shed light on us. KnowDDI indeed cannot impute missing edges as the relation type between two nodes disconnected in drug-flow subgraph is unknown. What KnowDDI can do is to add new edges representing resembling relationship and then propagate the learned drug similarities to implicitly compensate for the lack of known DDIs.

Therefore, in the revised manuscript, we add an edge of relation type r_{sim} which exists in external KG instead of r_{NA} between two nodes if they are estimated to be highly relevant. We estimate the relevance score $C_{h,t}^{(\tau)}(u, v, r)$ of relation r between any node-pair (u, v) in subgraph $S_{h,t}$ in equation (3). Particularly,

- If u and v are connected by relation r in $\bar{S}_{h,t}$, $C_{h,t}^{(\tau)}(u, v, r)$ represents the importance of a given edge. This edge will be dropped if $C_{h,t}^{(\tau)}(u, v, r)$ is small.
- While if u and v are disconnected in $\bar{S}_{h,t}$, meaning their connections are unknown, $C_{h,t}^{(\tau)}(u, v, r)$ represents the similarity between (u, v) . A new edge will be imputed if $C_{h,t}^{(\tau)}(u, v, r)$ is large.

In this way, the learned $S_{h,t}$ filters out information irrelevant to DDI prediction task and adds in resembling relationship between drugs.

With this change, we re-implement KnowDDI and redo all the experiments. In the revised manuscript, we carefully correct the statements about missing edges, revise Section 5.2.3 about adding new edge of relation type r_{sim} , and update the results reported by KnowDDI.

Q3. Fig.3(b-d) is not easy to understand. First, authors are supposed to claim that these three subfigures are presented for three samples, maybe you can draw horizontal lines instead of vertical lines to distinguish node pairs. The vertical lines are misleading. Second, the drug pair is supposed to be enlarged. Third, why are edges in the knowledge subgraph bidirectional? The importance of each edge is calculated based on the triple fact and the graph used in your work should be a directed graph.

Reply. Thanks for the suggestion. In Fig. 3 of the revised manuscript (reproduced as Fig.R1), we make the following changes: i) use horizontal lines to separate the subgraphs corresponding to different exemplar drug-pairs; ii) draw a red dashed circle encompassing the target drug-pair, denote head drug h by filled plus and tail drug t by star; iii) separately mark drugs from DDI graph in blue, drugs from external KG in gray, and other entities in green; iv) separately mark edges existing in DDI graph by solid blue lines, edges existing in external KG by solid gray lines, and new edges of type “resemble” estimated by KnowDDI by dotted gray lines. In submitted manuscript, new edge type r_{NA} has no direction in semantic meaning, we use bidirectional edges to represent the new edges. In the revised manuscript, we assign type r_{sim} to new edges, where r_{sim} exists in external KG. As the semantics of resemblance are not directional, we use undirected edges to represent relation type “resemble”.

Fig. R1: Comparing different subgraphs extraction strategy. **a**, F1 (%) obtained with different portion (%) of triplets sampled from the external KG on DrugBank. **b-e**, four exemplar drug-pairs (h, t)s in DrugBank and their enclosing subgraphs (first column), drug-flow subgraphs (second column), and knowledge subgraphs without propagating resembling relationship (third column) and knowledge subgraphs (fourth column). Drugs from DDI graph are marked in blue, drugs from external KG are colored in gray, while the other entities are colored in green. In particular, we draw a red dashed circle encompassing the target drug-pair, and mark the head drug h by filled plus and the tail drug t by star. We mark edges existing in DDI graph and external KG in solid blue lines and solid gray lines respectively. We specially use undirected edges to represent relation type “resemble” as the semantics of resemblance are not directional, and plot new edges of relation type “resemble” estimated by KnowDDI with dotted gray lines. The thickness of edges is used to reveal the magnitude of connection strength learned by KnowDDI.

Q4. In the section of “Embedding visualization”, Fig.4(a) is meaningless from my point of view. This example cannot show the advantage of KnowDDI nor its interpretability. The KGNN has the same ability or those spatial-based GNN methods can also learn different embeddings for drugs in two different triplets. Moreover, it has the same problem of Fig.3. The authors are supposed to enlarge the drug nodes.

Reply. *i) Differences between KnowDDI and other GNN methods.* Spatial-based GNN methods, e.g., GAT, GraphSAGE, and GCN, generate a unique embedding for each node in the combined network, then take embedding of drug i and drug j to make up the drug-pair embedding. Thus, different triplets can have distinctive embeddings, but the same drug in different triplets still has the same embedding. KGNN follows the same routine, while it uniformly samples a fixed number of nodes as neighbors of each node to reduce computational overhead. In the end, each drug still obtains a unique representation in KGNN. In contrast, methods which extract subgraphs distinctive to drug-pairs such as SumGNN and our proposed KnowDDI can obtain drug-pair-aware representations. The same drug in different triplet will have different representations. In particular, KnowDDI can obtain more distinctive drug-pair embeddings. In our Section 5.2.1, we first use a spatial-based GNN method, i.e., GraphSAGE, to obtain the generic node embeddings for nodes in the combine network. Then, we learn to remove irrelevant edges and add new edge of type “resemble” based on learned generic node embeddings. Upon this purified subgraph (i.e., knowledge subgraph) with respect to the target drug-pair, we transforms generic node embeddings to be distinctive to each drug-pair, which can be more predictive of DDI types. Please also check Q1 in general response where we discuss related works.

ii) Update Fig.4 . To demonstrate the advantage of KnowDDI clearly, we update Fig.4 (reproduced as Fig.R2) to plot the drug-pair embeddings instead. We also revise “embedding visualization” on page 10 as follows.

“ Finally, we wish to show that KnowDDI helps better shape the embeddings of drug-pairs and relations to be more predictive of DDI types between two drugs. From the DDI dataset, we randomly sample ten DDI relations, then randomly sample fifty triplets per relation. Given a drug-pair (h, t) , it is represented as the concatenation of drug embeddings which correspond to the node embeddings of h and t . For methods operating on subgraphs, the drug-pair embedding is obtained by concatenating drug embeddings with subgraph embedding which is obtained by mean pooling over node embeddings of all nodes in the subgraph [21,22]. In this way, local context of target drug-pair is leveraged to obtain better prediction results. We compare KnowDDI with SumGNN and KnowDDI (w/o resemble). In addition, we also show the drug-pair embeddings obtained by simply learning generic node embeddings without refining them on subgraphs.

Fig.4 shows the t-SNE visualization [34] obtained on DrugBank. First, we can see from generic embedding in Fig.4(b), to KnowDDI (w/o resemble) in Fig.4(c), to KnowDDI in Fig.4(d), drug-pair with the same relation type are getting closer while drug-pair with different relation types are becoming farther. Also, as can be seen, the clusters are more obvious in KnowDDI (Fig.4(c) and Fig.4(d)) than that of SumGNN (Fig.4(a)). This means that learning knowledge subgraphs is beneficial to obtain more distinctive drug-pair embeddings. ”

256
 257
 258
 259
 260
 261
 262
 263
 264
 265
 266
 267
 268
 269
 270
 271
 272
 273
 274
 275
 276
 277
 278
 279
 280
 281
 282
 283
 284
 285
 286
 287
 288
 289
 290
 291
 292
 293
 294
 295
 296
 297
 298
 299
 300
 301
 302
 303
 304
 305
 306

Fig. R2: T-SNE visualization of drug-pair embeddings. The four graphs share the same relations and drug-pairs. **a**, T-SNE of drug-pair embeddings obtained by SumGNN. **b**, T-SNE of drug-pair embeddings which are composed of concatenated generic node embeddings. **c**, T-SNE of drug-pair embeddings obtained by KnowDDI (w/o resemble). **d**, T-SNE of drug-pair embeddings obtained by KnowDDI.

Q5. How about removing the Generic node embedding generation? Is it necessary?

Reply. The design of generic node embedding is necessary. In KnowDDI, generic node embeddings are obtained by running a GNN on the combined network \mathcal{G} . In this way, we encode the generic knowledge of various other type of entities in \mathcal{G} and therefore enrich representation for drug nodes. The enriched drug representations and drug similarities propagated on the knowledge subgraph together implicitly compensate for the lack of DDIs. We update Fig.1 (reproduced as Fig.R3) to make the usage of “generic node embedding” more clear in the revised manuscript.

Fig. R3: Overview of KnowDDI. On a combined network which merges the DDI graph with an external KG, generic embeddings of all nodes are firstly learned to capture generic knowledge. Then for each target drug-pair, a drug-flow subgraph is extracted from the combined network, whose node embeddings are initialized as the generic embeddings. Via propagating the drug resembling relationship, the generic embeddings are transformed to be more predictive of the DDI types between the drug-pair, and the drug-flow subgraph is adapted as knowledge subgraph which contains explaining paths to interpret the prediction result.

The performance of without “Generic node embedding generation” can be seen from Fig.2(b) (reproduced as Fig.R4(b)) in the revised manuscript. It can be observed that the performance of KnowDDI gradually drop with less triplets from external KG being used. Without “Generic node embedding generation” is the same as the case of using 0% triplets. In this case, the performance deteriorates from 91.67% to 81.91%.

Q6. What is the effect of different lengths of the path on the learned embedding?

Reply. Please check Supplementary Fig.3(a) (reproduced as Fig.R5) of the revised manuscript, which is Supplementary Fig.2(a) in the submitted version. Recall that KnowDDI targets at selectively aggregating the global topology information from the combined network and the drug-pair-specific information encoded in the subgraph. As can be observed, neither too small nor too large P offers satisfactory performance. Therefore, one should start with a drug-flow subgraph with proper size by tuning P .

358
 359
 360
 361
 362
 363
 364
 365
 366
 367
 368
 369
 370
 371
 372
 373
 374
 375
 376
 377
 378
 379
 380
 381
 382
 383
 384
 385
 386
 387
 388
 389
 390
 391
 392
 393
 394
 395
 396
 397
 398
 399
 400
 401
 402
 403
 404
 405
 406
 407
 408

Fig. R4: The proposed KnowDDI is compared with (i) SumGNN which is a GNN-based method (w/ external KG) and obtains the second-best results in Table 1, (ii) SkipGNN which is the best among the GNN-based methods (w/o external KG) in Table 1, and (iii) KnowDDI (w/o resemble) which does not propagate resembling relationship between nodes whose connections are unknown. **a**, F1 (%) obtained for relations with different number of DDI fact triplets on DrugBank. **b**, F1 (%) obtained with different portion (%) of triplets sampled from the external KG on DrugBank. SkipGNN, which does not use external KG, is plotted just for reference.

Fig. R5: Varying the length P of relational path in KnowDDI on DrugBank.

Q7. Why do the authors integrate subgraph embedding into the calculation of $\hat{y}_{h,t}$? Please explain the rationality behind this integration.

Reply. Existing methods operating on subgraphs such as GraIL and SumGNN commonly concatenate subgraph embedding with drug embeddings to leverage local context to obtain better prediction results. We follow this line of research, so we also use subgraph embedding when calculating $\hat{y}_{h,t}$. In the revised manuscript, we mention it on page 10 and discuss it on page 12. In addition, we update Supplementary Section E.1 of the revised manuscript to include variant “w/o subgraph emb” which does not use subgraph embedding $\mathbf{h}_{S_{h,t}}$ to calculate $\hat{y}_{h,t}$. Table R2 shows the performance. We also update Supplementary Fig. 2 (reproduced as Fig.R6) in the revised manuscript. As can be seen, “w/o subgraph emb” performs worse, which validates the effectiveness of leveraging subgraph embedding to explicitly encode the local context of the knowledge subgraph.

Table R2: Performance (%) comparison between KnowDDI and w/o subgraph emb.

	AUROC	AUPRC	AP@50
w/o subgraph embedding	89.04±0.62	91.46±0.11	90.06±0.14
KnowDDI	91.53±0.24	93.17±0.09	91.89±0.11

Fig. R6: Different knowledge subgraph generation strategies for KnowDDI on DrugBank.

460 **Q8.** Some recently published papers that also make use of knowledge graph for DDI prediction are missed
 461 for discussion in related work, such as “Attention-based Knowledge Graph Representation Learning for
 462 Predicting Drug-drug Interactions”. DOI: 10.1093/bib/bbac140.

463 **Reply.** Thanks for the suggestion. In the revised manuscript, we discuss DDKG proposed in the sug-
 464 gested reference on Section 3.2 and Discussion section. DDKG improves upon KGNN. Recall that KGNN
 465 uniformly selects N nodes to form the neighborhood to propagate messages. DDKG improves KGNN
 466 by assigning attention weights to the N nodes during message passing, where the attention weights are
 467 obtained by calculating the similarity between initial node embeddings constructed from SMILES. Even-
 468 tually, DDKG obtains an unique presentation for each drug like KGNN. In contrast, KnowDDI obtains
 469 node-pair-aware representations by iteratively refining node embeddings and subgraphs. The generation
 470 of generic node embeddings and knowledge subgraphs also contribute to achieving the balance of generic
 471 information and drug-pair-aware local context during learning. We run DDKG using the public codes pro-
 472 vided by the respective authors and add its results obtained on two benchmark DDI datasets in Table 1
 473 in the revised manuscript (reproduced as Table R3). Empirical results validate our discussion, KnowDDI
 474 performs better than DDKG.
 475
 476

477 **Table R3:** Performance (%) comparison between KnowDDI and DDKG.

Dataset	DrugBank (multiclass)			TWO SIDES (multilabel)		
Metric	F1	ACC	Cohen’s κ	AUROC	AUPRC	AP@50
DDKG	75.84±0.22	88.70±0.39	87.53±0.21	93.15±0.18	91.09±0.39	87.50±0.43
KnowDDI	91.53±0.24	93.17±0.09	91.89±0.11	95.43±0.02	94.14±0.03	89.54±0.03

485
 486
 487
 488
 489
 490
 491
 492
 493
 494
 495
 496
 497
 498
 499
 500
 501
 502
 503
 504
 505
 506
 507
 508
 509
 510

To Reviewer #2

Q1. In the section of Abstract, the purpose of introducing KG is not to compensate for the lack of known DDIs, but to obtain the rich neighborhood information of each entity in KG

Reply. Thanks for the summary, we update Fig.1 (reproduced as Fig.R3) and revise the main text to make the following points clearer in the revised manuscript:

- The external KG which contains rich biomedicine and healthcare knowledge is used to enrich the drug representation. We first combine the provided DDI graph and the external KG into the combined network. Then, we obtain the generic embeddings for all nodes by running a GNN on the combined network.
- As similar drugs will have similar DDIs patterns, we learn knowledge subgraph for each drug-pair to propagate drug similarities between drug nodes.
- The enriched drug representations and the drug similarities propagated on the knowledge subgraph together implicitly compensate for the lack of known DDIs when learning a DDI predictor.

Q2. On line 101, it says “In addition, they are not developed for a particular downstream task such as DDI prediction considered in this paper, the mismatch of objectives can lead the deep learning model to learn irrelevant or distractive knowledge”. But both the methods of SumGNN [21] and KGNN [30] are constructed for the downstream task of DDI prediction. The method of KnowDDI in this manuscript is not the first one.

Reply. Yes, SumGNN and KGNN are both developed for DDI prediction. We never consider KnowDDI as the first deep learning method for DDI prediction. Precisely, we think KnowDDI firstly **adaptively leverage** the information of external KGs. Please note that on line 101 of the submitted version, “they” refer to external KGs. We mean that external KGs may contain irrelevant information for specific downstream applications. Thus, we design KnowDDI to adaptively leverage the information of external KG for the first time. Specifically, KnowDDI learns to drop existing edges if their estimated importance are low, or add edges of type “resemble” between disconnected nodes if their node embeddings are highly similar. After learning, KnowDDI obtains knowledge subgraphs which contain explaining paths pointing from head drug to tail drugs. We revise the paper to make these points clearer. Please also check Q1 in general response.

Q3. The section Dataset about data extract and description is missing in the manuscript

Reply. Please check Q2 in general response.

Q4. With the representation learned from the previous sections, how to predict the interaction value between drug-drug pairs, including the multi-class DDI prediction and the multi-label DDI prediction?

Reply. The prediction result is obtained by equation (7) in the manuscript. In the revised manuscript, the following inference steps are described in Section 5.2.4, and the details of testing procedure are offered in Supplementary Algorithm 3.

“ During inference, given a new drug-pair (h', t') where $h', t' \in \mathcal{V}$, we directly use KnowDDI with optimized θ_g, θ_k to obtain the class prediction vector $\hat{\mathbf{y}}_{h', t'}$. For multiclass prediction, the class is predicted as the relation which obtains the highest possibility in $\hat{\mathbf{y}}_{h', t'}$. As for multilabel prediction, the complete $\hat{\mathbf{y}}_{h', t'}$ is returned. Please refer to Supplementary Algorithm 3 for details. ”

Q5.1. In line 382, it uses the assumption of “similar drugs may have similar properties” to explain the interaction between Drug 103 and Drug 1127, which seems strained. Because it explains drug 103 is similar to drug 82, drug 82 is similar to drug 121, drug 121 is similar to drug Budesonide, drug 622 is similar to drug 1127, and there is no interaction between drug 121 and drug 622. Therefore, it is too strained to

562 explain the occurrence reason of DDI by so many similarity hops, which oppositely reflects the difference
 563 between drugs in many cases. It is best to find literature and database to support their conclusion.

564 **Reply.** Motivated by the common observation that similar drugs usually have similar DDIs patterns
 565 [r1,r2], we learn knowledge subgraph for each drug-pair to propagate drug similarities between drug
 566 nodes to implicitly compensate for the lack of known DDIs. In particular, we add new edges of rela-
 567 tion type “resemble” between nodes which are not connected in drug-flow subgraph but have highly
 568 similar node embeddings. In the case of drug-pair (103,1127), without the newly added edge of rela-
 569 tion type “resemble” pointing from 121 (Desonide) to 622 (Mestranol), the discovered explaining path
 570 “103 $\xrightarrow{\text{resembles}}$ 121 $\xrightarrow{\text{resembles (new edge)}}$ 622 $\xrightarrow{\text{resembles}}$ 1127” no longer exists. We also provide the molecu-
 571 lar graphs of drugs appearing in the explaining path in Fig. R7. As can be seen, these drugs have similar
 572 substructures which may explain the similarities.
 573

582 **Fig. R7:** Molecular graphs of drugs appearing in the explaining path “103 $\xrightarrow{\text{resembles}}$
 583 121 $\xrightarrow{\text{resembles (new edge)}}$ 622 $\xrightarrow{\text{resembles}}$ 1127” presented in Table 2 in the manuscript.
 584
 585

586 [r1] Vilar, S. et al. Drug-drug interaction through molecular structure similarity analysis. Journal of the
 587 American Medical Informatics Association 19 (6), 1066–1074 (2012)

588 [r2] Ding, H., Takigawa, I., Mamitsuka, H. & Zhu, S. Similarity-based machine learning methods for
 589 predicting drug–target interactions: A brief review. Briefings in Bioinformatics 15 (5), 734–747 (2014) .
 590

591 **Q5.2.** In addition, why is it that only drug nodes are found in the interpretable path, and there are no
 592 other entity nodes of the KG, such as proteins and diseases? Adding other entity nodes can mine more
 593 meaningful interpretability of DDI.
 594

595 **Reply.** The explaining paths in knowledge subgraph only contain drug nodes due to the design of drug-
 596 flow subgraph. Recall that knowledge subgraph is learned from drug-flow subgraph, containing the same
 597 set of nodes but the edges are adjusted during learning. In Section 5.2.2, we introduce drug-flow subgraph
 598 which only consists of relational path pointing from head drug to tail drug. In other words, nodes which
 599 are not on any relational paths pointing from head drug to tail drug are removed. While in external KG,
 600 which is Hetionet used in this paper, only drugs have incoming edges with drugs and the relation type is
 601 “Compound–resembles–Compound”. Thus, drug-flow subgraphs only contain drug nodes. In the revised
 602 manuscript, we mention this on page 10. We also replot Fig.1 (reproduced as Fig.R3) in the revised
 603 manuscript to highlight that drug-flow subgraphs only contain drug nodes.

604 In KnowDDI, other entity nodes are not used to raise interpretability. But they contribute to learning
 605 enriched drug representations, when we learn generic node embeddings from the combined network which
 606 merges DDI graph and external KG.
 607
 608
 609
 610
 611
 612

To Reviewer #3

Q1. As the authors claim, KnowDDI is the first deep method exploiting biomedical KGs for DDI prediction. They make dedicated designs to transform the noisy biomedical KGs into operatable knowledge graphs for the DDI prediction. The novelty could be enhanced with detailed discussion about KnowDDI’s connections to previous GNN-based DDI methods with external KG.

Reply. Thanks for the suggestion. Please check Q1 in general response.

Q2.1 As the transformation from external KGs to operatable graphs lies in the core of the whole method, related formulation and statement could be more detailed.

Reply. We merge the provided DDI dataset and external KG into a combined network, whose formulations are provided in Section 5.1. The procedures of empirically separately merging the two benchmark DDI datasets DrugBank and TWOSIDES with external KG Hetionet are described in Section 3.1 of the revised manuscript. Besides, the procedure of extracting drug-flow subgraphs from the combined network is summarized in Supplementary Algorithm 1.

Q2.2 Specifically, more introduction about the MLP adopted to estimate relevance scores of relations would facilitate the illustration of the refining process, along with its role in the entire method.

Reply. Thanks for the suggestion. In Section 5.2.3, we learn a knowledge subgraph from generic embeddings and the drug-flow subgraph. During this process, irrelevant edges are removed and new edges of type “resemble” are added between nodes with highly similar node embeddings. To estimate the relevance between any two nodes on the subgraph, we use MLP to estimate the relevance score $\mathbf{C}_{h,t}^{(\tau)}(u, v, r)$ for each relation $r \in \mathcal{R}_{h,t}$ between a node-pair (u, v) during knowledge subgraph generation. A large $\mathbf{C}_{h,t}^{(\tau)}(u, v, r)$ reveals that (u, v) is highly relevant in terms of r . In the revised manuscript, we replot Fig.1 (reproduced as Fig.R3) and mark the subsection titles of Method section. Thus, the role of knowledge subgraph generation (refining of subgraph structure and node embeddings) can be clearer. We also add a summary of hyperparameters in Supplementary Table 3, which provides the detailed architectures of MLP.

Q2.3 Several notations are used without clarification, e.g., $AGG()$ in Eq.1, $COMB()$ in Eq.2.

Reply. Thanks for the reminder. In the submitted manuscript, $AGG()$ in equation (1) stands for aggregation function, while $COMB()$ equation (1) stands for combination functions. We now replace them by exact formulations. Please see equation (1) and equation (2) in the revised manuscript. We also carefully check the use of all notations.

613
614
615
616
617
618
619
620
621
622
623
624
625
626
627
628
629
630
631
632
633
634
635
636
637
638
639
640
641
642
643
644
645
646
647
648
649
650
651
652
653
654
655
656
657
658
659
660
661
662
663

To Reviewer #4

Q.1.1. The only difference between the proposed method and SumGNN is the missing edge imputation during the knowledge subgraph generation, which has been widely studied in knowledge graph completion.

Reply. In the revised manuscript, we highlight that KnowDDI does not target at imputing missing edges. It learns new edges of type “resemble” to propagate drug similarities. In comparison to SumGNN, KnowDDI implicitly compensates for the lack of DDIs by enriched drug representations and propagated drug similarities. We add the following discussion comparing KnowDDI to SumGNN in the second paragraph of Discussion section.

“While SumGNN extracts the enclosing subgraphs of the drug-pair, then prunes edges based on the node features. By encoding local context within the subgraph, these methods obtain node-pair-aware representations, i.e., a drug can be represented different depending on which drug to interact. Our KnowDDI also learns with subgraphs, while two major design differences makes it obtain better and more interpretable performance. The first difference is that KnowDDI first learns generic node embeddings on the combined network which enriches the drug representation, then transforms them on knowledge subgraph to incorporate with the local context of drug-pair. The second difference is the adjustment of subgraph where existing edges can be dropped if their estimated importance are low, and new edges of type “resemble” can be added between disconnected nodes if their node embeddings are highly similar. This allows KnowDDI to capture explaining paths pointing from head drug to tail drug. While SumGNN directly learns drug-pair-aware representations from the extracted subgraph. With the two differences, KnowDDI achieves the balance of generic information and drug-pair-aware local context during learning.”

Q.1.2. An ablation study of comparing the model w/o missing edge imputation (only keeping irrelevant edge pruning) and SumGNN is required to demonstrate the model performance.

Reply. In the revised manuscript, we consider KnowDDI (w/o resemble), a variant of KnowDDI which does not add new edges of type “resemble” between nodes with highly similar node embeddings and keeps irrelevant edge pruning. The relation-wise performance in Fig.2(a) (reproduced as Fig.R4(a)) in the revised manuscript consistently show that KnowDDI outperforms KnowDDI (w/o resemble). In addition, Fig.2(b) (reproduced as Fig.R4(b)) in the revised manuscript, which plots the performance changes w.r.t. varying portion (%) of triplets sampled from the external KG, shows that the performance gap between KnowDDI (w/o resemble) and KnowDDI gets larger with fewer triples. This means drugs similarities have stronger influence on compensating for the lack of DDIs when drug representations are less enriched by external KG.

Q.1.3. Even though, the results in Fig.3(a) show that the performance of the model w/ knowledge subgraph is comparable to the model w/ drug pair-aware subgraph (i.e., < 1% improvement)

Reply. In Fig. 3(a) of submitted manuscript, the performance gain reported by subgraph learning is not huge because generic embedding can offer rich information which compensates for the lack of known DDIs. In the revised manuscript, we update Fig.3(a) (reproduced as Fig.R1(a)) to plot the results obtained by KnowDDI with various subgraphs on DrugBank given different percentages of triplets from external KG. We also revise observations on page 7 accordingly.

“...by comparing the performance of drug-flow subgraph and knowledge subgraph under different percentages of triplets, we can see that both enriched drug representations and propagated drug similarities contribute to the performance improvements. However, the performance gain is larger when fewer triplets are used. This means removing irrelevant edges and propagating drug similarities play a stronger influence on compensating for the lack of DDIs when drug representations are less enriched. In summary, learning knowledge subgraph is effective.”

Q2. In Table 1, the performance improvement over the best baseline in TWOSIDES is relatively small (i.e., 1%). Given this limited performance gain, it’s doubtful that the proposed method can robustly and consistently achieve good performance on other different datasets.

Reply. Please note that the benchmark TWOSIDES used in this study contains 200 commonly occurring relations, where each of them has at least 900 fact triplets. In the revised manuscript, we highlight it in Section 3.1.

Fig. R8(a) and Fig. R8(b) show statistics of relation frequency for DrugBank and TWOSIDES. As can be seen, relations in DrugBank follow long-tail distribution, while relations in TWOSIDES have more comparable fact triples. Thus, it can be easier to learn DDI predictor on TWOSIDES, where our KnowDDI still performs the best.

Fig. R8: Statistics of relation frequency for DrugBank (a), TWOSIDES (b), and TWOSIDES-Large (c). Each bin represents the performance improvement of one relation. From leftside to rightside of x-axis, relations are ordered by decreasing number of associated fact triplets.

To validate the general effectiveness of KnowDDI, we construct a new dataset called TWOSIDES-Large. In particular, we rank relations by decreasing number of associating fact triplets, and choose the 900 relations ranked between 300 to 1200. Fig. R8(c) shows statistics of relation frequency for TWOSIDES-Large. As shown, relations in TWOSIDES-Large contain varying number of fact triplets. Table R4 shows the performance of KnowDDI and the second-best method SumGNN obtained on TWOSIDES-Large. As expected, on this large dataset with both sample-sufficient and rare relations, the performance gain of KnowDDI over SumGNN is large. Please also see our reply to Q.4.1. where we provide relation-wise performance improvement of KnowDDI over SumGNN on DrugBank, TWOSIDES and TWOSIDES-Large. Results consistently shows KnowDDI obtains better performance.

766 **Table R4:** Performance (%) obtained by KnowDDI and the second-best SumGNN on TWOSIDES-Large.
 767

	AUROC	AUPRC	AP@50
SumGNN	93.10±0.09	90.95±0.21	85.22±0.37
KnowDDI	95.40±0.06	94.14±0.12	89.60±0.02

773 **Q3.** Missing baselines. Some related work in DDI prediction with biomedical KG is neither cited nor
 774 compared in the paper.

- 775 1. Hong, Yue, et al. "LaGAT: link-aware graph attention network for drug–drug interaction prediction."
 776 Bioinformatics 38.24 (2022): 5406-5412.
- 777 2. Ren, Zhong-Hao, et al. "A biomedical knowledge graph-based method for drug–drug interactions
 778 prediction through combining local and global features with deep neural networks." Briefings in
 779 Bioinformatics 23.5 (2022).

781 **Reply.** Thanks for the suggestion. In the revised manuscript, we discuss LaGAT proposed in the first
 782 reference on Section 3.2 and Discussion section. Both LaGAT and KnowDDI learn with subgraphs and
 783 obtain drug-pair-aware representations. In particular, LaGAT extracts a subgraph consisting of a fixed
 784 number of nodes around the head and tail drugs, updates node embeddings by aggregating neighboring
 785 nodes based on attention weights calculated using node embeddings, and leaves the subgraph unchanged.
 786 In contrast, our KnowDDI makes two major design differences which lead to better and more interpretable
 787 performance. The first difference is the learning of generic node embeddings on the combined network
 788 which enriches the drug representation. The second difference is the adjustment of subgraph where existing
 789 edges can be dropped if their estimated importance are low, and new edges of type "resemble" can be
 790 added between disconnected nodes if their node embeddings are highly similar. With the two differences,
 791 KnowDDI achieves the balance of generic information and drug-pair-aware local context during learning.
 792 We run LaGAT using the public codes provided by the respective authors and add its results obtained
 793 on two benchmark DDI datasets in Table 1 (reproduced as Table R5) in the revised manuscript. As can
 794 be observed, our proposed KnowDDI outperforms LaGAT.

795
 796
 797 **Table R5:** Performance (%) comparison between KnowDDI and LaGAT.

Dataset	DrugBank (multiclass)			TWOSIDES (multilabel)		
Metric	F1	ACC	Cohen's κ	AUROC	AUPRC	AP@50
LaGAT	83.69±0.74	88.86±0.12	87.33±0.14	88.72±0.22	84.03±0.43	82.46±0.41
KnowDDI	91.53±0.24	93.17±0.09	91.89±0.11	95.43±0.02	94.14±0.03	89.54±0.03

805
 806 The second suggested reference focuses on fusing representations learned using SMILES, BioKG and
 807 biological function, which is disjoint with our research target. Besides, this reference reports that ComplEx
 808 obtains the best performance. While our baseline MSTE [15] already outperforms ComplEx. Hence, we
 809 do not compare with this reference.

810
 811 **Q.4.1.** The scores in Table 1 are not clinically meaningful - DDI prediction performance for each class
 812 (86 in DrugBank and 200 in TWOSIDES) is necessary to be included. More discussions of the model
 813 performance with respect to each class of DDI are expected, especially whether the model can achieve
 814 consistent performance in those relatively rare interactions.

815
 816

Reply. Table 1 in the manuscript summarizes the performance obtained by various baselines on the two benchmark datasets. Please note that Fig.2(a) in the submitted manuscript shows the relation-wise F1 score (%) grouped into bins according to the number of edges associated with the reaction. We compare the proposed KnowDDI with SkipGNN which performs the best among GNN-based methods (w/o external KG), and SumGNN which obtains the second-best among GNN-based methods (w/ external KG). Results show that our KnowDDI consistently outperforms the others for all relation groups.

To see the relation-wise performance with more details, we further plot the performance improvement of KnowDDI over SumGNN (the second-best method) on DrugBank, TWOSIDES, and TWOSIDES-Large in Fig. R9. As can be seen, KnowDDI obtains better performance than SumGNN on most relations, including those rare relations.

Fig. R9: Performance improvement of KnowDDI over SumGNN (the second-best method) on DrugBank (a), TWOSIDES (b), and TWOSIDES-Large (c). Each bin represents the performance improvement of one relation. From leftside to rightside of x-axis, relations are ordered by decreasing number of associated fact triplets. A higher (lower) bin suggests a larger performance improvement (decrease) for KnowDDI compared to SumGNN.

Q.4.2. Also, model performance on some important and commonly studied adverse drug reactions (e.g., acute liver failure, acute myocardial infarction, acute renal failure and upper GI bleeding) needs to be discussed.

Reply. Please note that there are no adverse drug reaction whose names exactly match the ones suggested. Instead, we manage to find adverse drug reactions with similar semantic meaning in TWOSIDES-Large. Table R6 summarizes their relation-wise performance, and the improvement of KnowDDI over SumGNN

868 is reported. Results show that our KnowDDI performs better than the second-best SumGNN on these
 869 commonly studied adverse drug reactions. Besides, as shown in Fig. R9(c), KnowDDI obtains improved
 870 performance for relations in TWOSIDES-Large in general.

871

872

873 **Table R6:** Performance comparison between KnowDDI and SumGNN on TWOSIDES dataset. The
 874 metric is AUROC (%).

875

876

877

878

879

880

881

882

883

884

885

886

887

888

889

890

891

892

893

894

895

896

897

898 **Q5.** The code and data are not provided for review.

899

900

901

902

903

904

905

906

907

908

909

910

911

912

913

914

915

916

917

918

Adverse Drug Reaction	KnowDDI	SumGNN	Improvement
liver abscess	96.16	93.34	2.82
myocarditis	96.52	94.54	1.98
adrenal insufficiency	95.68	93.56	2.12
hepatorenal syndrome	95.27	91.78	3.49
renal mass	95.36	94.10	1.26
renal tubular acidosis	92.24	87.92	4.32
renal colic	98.28	95.50	2.78
renal abscess	96.98	89.25	7.73
renal cancer	99.89	95.89	4.00
upper GI bleeding	93.75	91.03	2.72
lower GI bleeding	94.96	92.57	2.39

909 **Reply.** Please check Q3 in general response.

Reviewers' comments:

Reviewer #1 (Remarks to the Author):

The authors have made substantial improvements in this revision. However, a few issues remain.

1. According to the pipeline, the graph is constructed by combining the DDI graph and KG. I'm curious whether the authors have excluded all interactions from the training set. It's well-known that both KGNN and DDKG predict DDIs relying on the knowledge graph devoid of any DDIs. Furthermore, could you provide insight into the rationale behind integrating the DDI graph into the KG?

2. In this work, the authors develop a framework for multilabel DDI predictions. However, as a multilabel task, is it imperative to generate negative samples? Does this practice contribute to enhancing model performance or just following previous work?

3. There are still some grammar mistakes, please revise them. For example, in Abstract, drug representation -> drug representations, Line 634, denote -> denotes

Reviewer #2 (Remarks to the Author):

The authors have addressed all my previous comments. In this round, I only suggest that they validate the proposed approach in finding unknown DDIs excluded from their collected datasets.

Reviewer #3 (Remarks to the Author):

Thanks for the rebuttal from the authors. It does address my concerns. My comment on the paper remains unchanged.

Reviewer #4 (Remarks to the Author):

Thank you for the responses; they addressed most of my questions. However, I still have one additional follow-up question regarding the authors' reply to Q.4.2, as follows:

The authors stated that they could not find the mentioned ADRs in TWOSIDES. However, these ADRs can be defined using MedDRA codes and are available in TWOSIDES. For example, the MedDRA code for Acute myocardial infarction is 10000891 (<https://bioportal.bioontology.org/ontologies/MEDDRA?p=classes&conceptid=10000891>), and the MedDRA code for Acute kidney injury (renal failure) is 10069339 (<https://bioportal.bioontology.org/ontologies/MEDDRA?p=classes&conceptid=10069339>). These ADRs are important and widely studied in numerous drug safety literature. It is crucial to evaluate the model's performance against these representative ADRs instead of just showing an overall metric.

Rebuttal letter for “Accurate and interpretable drug-drug interaction prediction enabled by knowledge subgraph learning”

We thank all reviewers for their valuable suggestions. In this rebuttal letter, we provide point-by-point response to remaining questions. The reviewers’ comments are in blue, and our replies are in black.

To Reviewer #1

Q1.1. According to the pipeline, the graph is constructed by combining the DDI graph and KG. I’m curious whether the authors have excluded all interactions from the training set. It’s well-known that both KGNN and DDKG predict DDIs relying on the knowledge graph devoid of any DDIs.

Reply. We did not exclude interactions from the training set. During training, all fact triplets in training set are provided in the combined network. In contrast, KGNN and DDKG do not merge DDI graph and external KG. For each drug in target drug-pair, they extract its local neighborhood only from KG. Please check Data preprocessing section on page 3 where we describe the procedure to merge DDI graph and external KG into a combined network.

“ We formulate the benchmark DDI datasets as DDI graphs separately, whose statistics are summarized in Table 1. The fact triplets in DDI datasets are split into training, validation, and testing sets with a ratio of 7:1:2 following SumGNN [22] for fair comparison. We remove from external KG the drug-drug edges contained in DDI graph to avoid information leakage, then merge the resultant external KG and DDI graph into a large combined network. Eventually, the DDI graph of DrugBank is merged with a graph of 33765 nodes and 1690693 edges extracted from Hetionet, and the DDI graph of TWOSIDES is merged with a graph of 28132 nodes and 1666632 edges extracted from Hetionet, respectively. During training, the drug-drug edges in validation and testing sets are unseen. After tuning hyperparameters on fact triplets in validation set, the model performance is evaluated on fact triplets in testing set. ”

Q1.2. Furthermore, could you provide insight into the rationale behind integrating the DDI graph into the KG?

Reply. We merge the DDI graph with the external KG as a combined network. As illustrated in Fig. 1, we run a GNN on the combined network to obtain generic embedding of each node, which encodes knowledge from both DDI graph and external KG. These generic embeddings then serve as good start for subsequent refinement on knowledge subgraphs. In addition, viewing DDI graph and external KG as a complete unit enables us to find explaining paths pointing from head drugs to tail drugs, which can interpret the prediction results. Please check Interpretability section where we provide knowledge subgraphs learned from the combined network (Fig. 3) and explaining paths (Table 2) for exemplar drug-pairs.

Q2. In this work, the authors develop a framework for multilabel DDI predictions. However, as a multilabel task, is it imperative to generate negative samples? Does this practice contribute to enhancing model performance or just following previous work?

Reply. It is not imperative to generate negative samples. One can also conduct data augmentation, modify loss functions and re-weight labels to handle the label imbalance problem. For a fair comparison, we generate negative samples following GraIL [21] and SumGNN [22], as written on line 731, page 15. It is not the reason why our KnowDDI performs the best.

Q3. There are still some grammar mistakes, please revise them. For example, in Abstract, drug representation → drug representations, Line 634, denote → denotes

052 **Reply.** Please note that in sentence “Let $e_v^{(0)}$ denote the feature of node $v \in \mathcal{V}$ ” on line 634, we use
 053 ‘denote’ instead of “denotes” due to the use of “Let” at the beginning. We have carefully proofread the
 054 paper to remove possible grammar issues. Thanks for the suggestions!

055 056 To Reviewer #2

057
 058 **Q1.** The authors have addressed all my previous comments. In this round, I only suggest that they
 059 validate the proposed approach in finding unknown DDIs excluded from their collected datasets.

060 **Reply.** Thanks for recognizing our efforts! Please note that we evaluate model performance on testing
 061 DDI fact triplets which are unseen during training. If a new drug with a few associating fact triplets
 062 appears, we can merge the new drug and its fact triplets into the combined network, then apply KnowDDI.
 063 As for the appearance of new relations, we leave it as future work.

065 To Reviewer #3

066
 067 **Q1.** Thanks for the rebuttal from the authors. It does address my concerns. My comment on the paper
 068 remains unchanged.

069 **Reply.** Thanks for the positive comments and valuable time spent on reviewing our paper!

071 To Reviewer #4

072
 073 **Q1** Thank you for the responses; they addressed most of my questions. However, I still have one additional
 074 follow-up question regarding the authors’ reply to Q.4.2, as follows: The authors stated that they could
 075 not find the mentioned ADRs in TWOSIDES. However, these ADRs can be defined using MedDRA
 076 codes and are available in TWOSIDES. For example, the MedDRA code for Acute myocardial infarction
 077 is 10000891¹, and the MedDRA code for Acute kidney injury (renal failure) is 10069339². These ADRs
 078 are important and widely studied in numerous drug safety literature. It is crucial to evaluate the model’s
 079 performance against these representative ADRs instead of just showing an overall metric.

080
 081 **Reply.** Thanks for recognizing our efforts! We indeed agree that relation-wise performance, particularly
 082 performance on commonly studied ADRs, are important. While we use the overall metric to show that
 083 KnowDDI performs better in general. Due to the numerous classes, reporting performance on a per-
 084 relation basis in main text is space-limited. We now include a comparison of relation-wise performance
 085 in Supplementary section C (reproduced below), where performance on commonly studied ADRs are
 086 emphasized.

087
 088 “ Here, we first report the relation frequency. Supplementary Fig. R1(a) and Supplementary
 089 Fig. R1(b) show statistics of relation frequency for DrugBank and TWOSIDES. As can be seen, rela-
 090 tions in DrugBank follow long-tail distribution, while relations in TWOSIDES are associated with
 091 more comparable fact triples. Recall that TWOSIDES used in this study contains 200 commonly
 092 occurring relations with more than 900 fact triplets following SumGNN [22]. This explains why it is
 093 easier to predict DDI on TWOSIDES, as can be seen in Table 1. To validate the general effective-
 094 ness of KnowDDI, we use the original TWOSIDES [24] which contains 4649441 fact triplets for 645
 095 drugs and 1317 relations, and remove relations with less than 3 fact triplets. Afterwards, we obtain
 096 a new dataset denoted as TWOSIDES-Large, which contains 4649430 fact triplets for 645 drugs and
 097 1311 relations. We preprocess it as described in Data preprocessing section. Supplementary Fig. R1(c)
 098 shows statistics of relation frequency for TWOSIDES-Large. As shown, relations in TWOSIDES-
 099 Large contain varying number of fact triplets. ”

101 ¹<https://biportal.bioontology.org/ontologies/MEDDRA?p=classes&conceptid=10000891>

102 ²<https://biportal.bioontology.org/ontologies/MEDDRA?p=classes&conceptid=10069339>

Fig. R1: Statistics of relation frequency for DrugBank (a), TWOSIDES (b), and TWOSIDES-Large (c). Each bin represents the performance improvement of one relation. From leftside to rightside of x-axis, relations are ordered by decreasing number of associated fact triplets.

“Next, we provide relation-wise performance improvement of our KnowDDI over SumGNN which obtains the second-best performance in Table 1. Supplementary Fig.R2 shows the results obtained on DrugBank, TWOSIDES, and TWOSIDES-Large. As can be seen, KnowDDI obtains better performance than SumGNN on most relations, including those rare relations. As expected, KnowDDI demonstrates a larger performance gain over SumGNN on TWOSIDES-Large, which contains both sample-sufficient and rare relations, compared to TWOSIDES, which is more balanced in its composition.”

“Finally, Supplementary Table R1 shows the performance of KnowDDI and the second-best method SumGNN obtained on some important and commonly studied adverse drug reactions (ADRs) which exist in TWOSIDES-Large. Results show that KnowDDI consistently performs better on these relations.”

“In conclusion, KnowDDI not only excels in terms of overall metrics but also adeptly manages most relations, including those commonly studied. With our open-source approach, we anticipate that KnowDDI can contribute to predicting DDIs for other intriguing relations as well.”

154
155
156
157
158
159
160
161
162
163
164
165
166
167
168
169
170
171
172
173
174
175
176
177
178
179
180
181
182
183
184
185
186
187
188
189
190
191
192
193
194
195
196
197
198
199
200
201
202
203
204

Fig. R2: Performance improvement of KnowDDI over SumGNN (the second-best method) on DrugBank (a), TWOSIDES (b), and TWOSIDES-Large (c). Each bin represents the performance improvement of one relation. From leftside to rightside of x-axis, relations are ordered by decreasing number of associated fact triplets. A higher (lower) bin suggests a larger performance improvement (decrease) for KnowDDI compared to SumGNN.

Table R1: Performance comparison between KnowDDI and SumGNN on the representative ADRs from the complete TWOSIDES dataset. The metric is AUROC (%).

Adverse Drug Reaction	Acute Liver Failure	Acute Myocardial Infarction	Acute Renal Failure	Upper GI Bleeding
KnowDDI	92.23	93.15	90.74	94.78
SumGNN	90.83	91.95	88.90	92.87
Improvement	1.40	1.20	1.84	1.91